# Absence of TGFβ signaling in retinal microglia induces retinal degeneration and exacerbates choroidal neovascularization

Wenxin Ma[1], Sean M Silverman[1], Lian Zhao[1], Rafael Villasmil[2], Maria M Campos[3], Juan Amaral[4], Wai T Wong[1]*

[1]Unit on Neuron-Glia Interactions in Retinal Disease, National Eye Institute, National Institutes of Health, Bethesda, United States; [2]Flow Cytometry Core Facility, National Eye Institute, National Institutes of Health, Bethesda, United States; [3]Section on Histopathology, National Eye Institute, National Institutes of Health, Bethesda, United States; [4]Unit on Ocular Stem Cell and Translational Research, National Eye Institute, National Institutes of Health, Bethesda, United States

**Abstract** Constitutive TGFβ signaling is important in maintaining retinal neurons and blood vessels and is a factor contributing to the risk for age-related macular degeneration (AMD), a retinal disease involving neurodegeneration and microglial activation. How TGFβ signaling to microglia influences pathological retinal neuroinflammation is unclear. We discovered that ablation of the TGFβ receptor, TGFBR2, in retinal microglia of adult mice induced abnormal microglial numbers, distribution, morphology, and activation status, and promoted a pathological microglial gene expression profile. TGFBR2-deficient retinal microglia induced secondary gliotic changes in Müller cells, neuronal apoptosis, and decreased light-evoked retinal function reflecting abnormal synaptic transmission. While retinal vasculature was unaffected, TGFBR2-deficient microglia demonstrated exaggerated responses to laser-induced injury that was associated with increased choroidal neovascularization, a hallmark of advanced exudative AMD. These findings demonstrate that deficiencies in TGFβ-mediated microglial regulation can drive neuroinflammatory contributions to AMD-related neurodegeneration and neovascularization, highlighting TGFβ signaling as a potential therapeutic target.
DOI: https://doi.org/10.7554/eLife.42049.001

*For correspondence:
wongw@nei.nih.gov

Competing interests: The authors declare that no competing interests exist.

## Introduction

The development of neuroinflammatory changes in the retina is a significant factor in the pathogenesis of multiple retinal disorders including glaucoma (*Williams et al., 2017*), diabetic retinopathy (*Xu and Chen, 2017*), and age-related macular degeneration (*Guillonneau et al., 2017*). Abnormal immune responses arising from physiological changes in microglia, the primary resident innate immune cell in the retina, are thought to drive aspects of disease progression, including neuronal degeneration and pathological neovascularization (*Karlstetter et al., 2015*; *Silverman and Wong, 2018*). Under healthy conditions, microglia in the retina integrate a variety of constitutive regulatory signals from other neighboring cells (*Fontainhas et al., 2011*; *Liang et al., 2009*), enabling them to perform homeostatic roles in maintaining retinal structure and function (*Wang et al., 2016*). How retinal microglia transition from a homeostatic physiological state to ones that promote disease progression is however not well understood. Elucidation of molecular mechanisms governing these

transitions is likely central to designing strategies for microglial modulation in retinal disease (*Arroba and Valverde, 2017*; *Bell et al., 2018*).

TGFβ signaling is a significant influence on the regulation of microglial development and mature function in the brain *in vivo* (*Butovsky et al., 2014*; *Buttgereit et al., 2016*) and promoting microglial survival and specification *in vitro* (*Bohlen et al., 2017*). Altered TGFβ signaling in microglia has been linked to pathogenic mechanisms of neurodegenerative disorders in the brain and spinal cord (*Lund et al., 2018a*; *Taylor et al., 2017*). In the retina, TGFβ signaling exerts pleotropic effects on multiple retinal cell types that underlie numerous functions ranging from maintaining retinal neuronal differentiation and survival (*Braunger et al., 2013*; *Walshe et al., 2011*) to regulating the development and structural integrity of retinal vessels (*Braunger et al., 2015*; *Walshe et al., 2009*). However, the specific role of TGFβ signaling to retinal microglia in the regulation of homeostatic vs. pathologic states, and how this may contribute to retinal disease pathogenesis, are not known. Significantly, TGFβ signaling has been implicated in the pathobiology of age-related macular degeneration (AMD), the leading cause of vision loss in older patients in the developed world (*Jager et al., 2008*) and a condition still lacking comprehension prevention and treatment. Alterations in the levels of TGFβ ligands have been reported in eyes of AMD patients (*Tosi et al., 2017*; *Tosi et al., 2018*). Genome-wide association studies have discovered that polymorphisms in TGFBR1, a receptor transducing TGFβ signals in conjunction with TGFBR2, influence the risk for developing AMD (*Fan et al., 2017*; *Fritsche et al., 2013*). HTRA1, another significant AMD risk-associated protein, has been thought to confer increased AMD risk by differentially binding to and cleaving intraocular TGFβ−1, altering TGFβ signaling to microglia (*Friedrich et al., 2015*). These findings have prompted the consideration of TGFβ signaling as a potential target for AMD therapy (*Fisichella et al., 2016*; *Platania et al., 2017*). However, how direct TGFβ signaling regulates microglial physiology in the retina to influence inflammatory, neurodegenerative, and neovascular processes in AMD is not elucidated.

Here, we investigate the role of direct and constitutive TGFβ signaling to microglia by inducing microglia-specific ablation of TGFBR2, a receptor required for TGFβ signal transduction, in the adult mouse retina. We found that inhibition of TGFβ signaling in microglia induced abnormalities in microglial homeostasis in the retina, altering overall microglial number, distribution, and morphology. These changes resulted in reduced physical coverage of the retinal plexiform layers by microglial processes, and likely diminished microglial trophic support. TGFβ signaling ablation resulted in a downregulation of microglial 'sensome' genes and an upregulation of microglial activation markers. These microglial changes were highly consequential to the maintenance of a healthy retina, inducing widespread Müller cell gliosis and structural and functional degeneration of retinal neurons. Retinal microglia deficient in TGFβ signaling also demonstrated abnormal injury responses that promoted increased choroidal neovascularization in a laser-induced model of injury. Taken together, our findings indicate that constitutive neuron-microglia interactions in the form of TGFβ signaling are necessary in the maintenance of the orderly organization and trophic function of microglia in the retina; in its absence, microglia undergo pathologic transformation in ways that promote retinal changes resembling those observed in AMD pathology. These results provide insight into how abnormal TGFβ signaling in retinal microglia can contribute causally to AMD pathobiology, and raise the possibility that microglia may be modulated via TGFβ signaling as a potential therapeutic strategy.

## Results

### Constitutive TGFBR2 expression in microglia of the adult mouse retina is specifically ablated in *Cx3cr1$^{CreER/+}$, Tgfbr2$^{flox/flox}$* (TG) mice

We characterized TGFBR2 expression by performing immunohistochemical analysis in flat-mounted retina from two-month old adult *Cx3cr1$^{+/GFP}$* mice. CX3CR1-expressing microglia in both the inner and outer plexiform layers (IPL, OPL) demonstrated immunopositivity for TGFBR2 in the cell membranes of somata and ramified processes (*Figure 1A*), as did CD31+ retinal endothelial cells. These findings were in agreement with RNAseq mRNA expression profiles in specific cells types; in the adult mouse retina (*Siegert et al., 2012*), *Tgfbr2* mRNA expression was high in microglia but low in other retinal neurons (*Figure 1B*), while in the postnatal mouse brain (*Bennett et al., 2016*), *Tgfbr2*

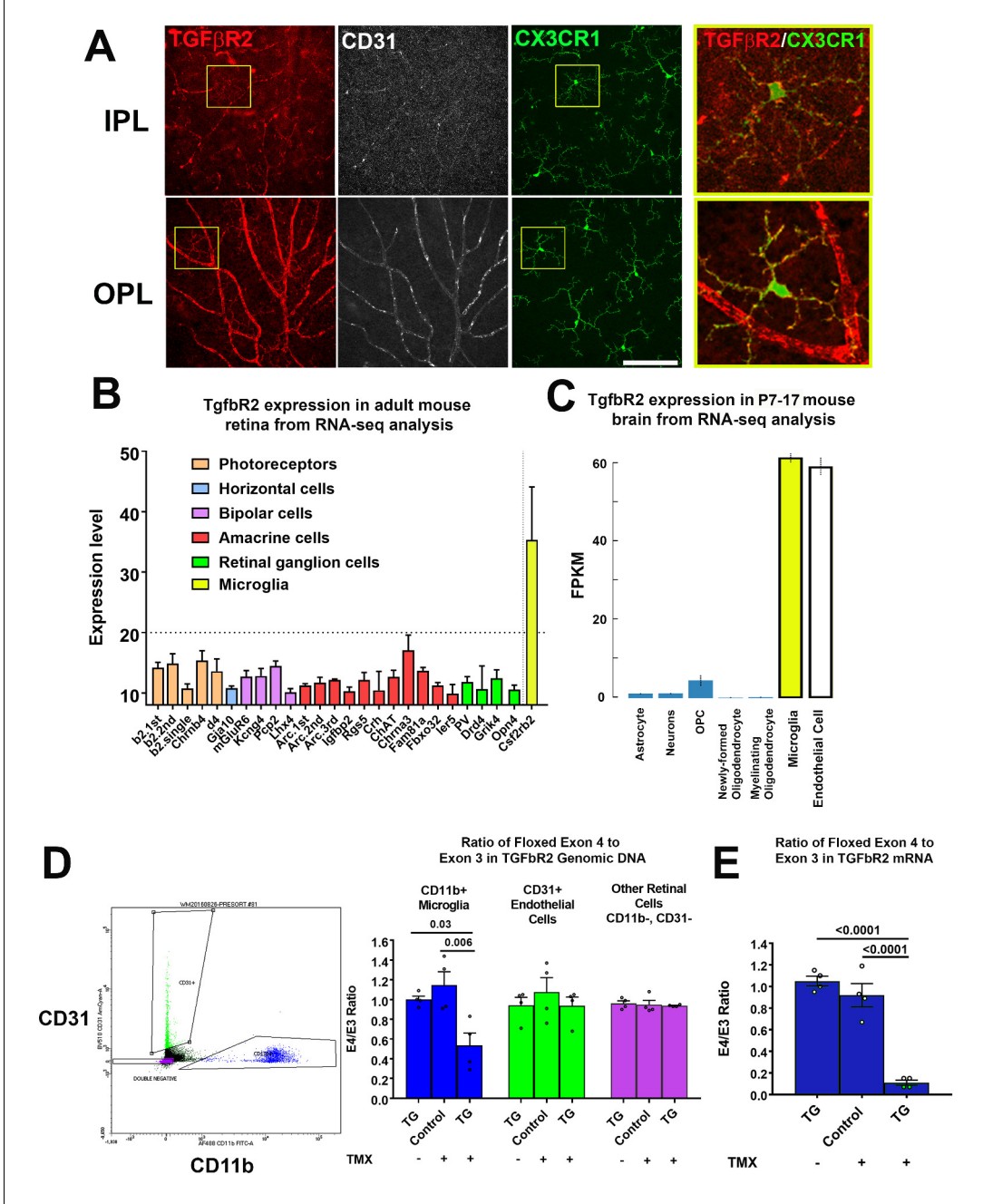

**Figure 1.** TGFBR2 is constitutively expressed in adult mouse retinal microglia and is specifically ablated in retinal microglia of adult *Cx3Cr1$^{CreER/+}$*, *Tgfbr2$^{flox/flox}$* (TG) mice upon tamoxifen induction. (**A**) Immunohistochemical labeling for TGFBR2 (*red*) in the adult CX3CR1$^{+/GFP}$ mouse retina was localized prominently to CD31-immunopositive vascular endothelial cells (*white*) and to CX3CR1-expressing, GFP+ microglia cells (*green*) in both the inner and outer plexiform layers (IPL, OPL). Insets (*yellow boxes*) show microglia demonstrating colocalization of microglial marker CX3CR1 with TGFBR2. Scale bar = 100 μm. (**B**) Reference to an atlas of specific cell type transcriptomes from the adult mouse retina highlighted constitutive expression of *Tgfbr2* mRNA in retinal microglia, with very low or no expression in different classes of retinal neurons (expression levels >20 correspond to significant expression). (**C**) Reference to an atlas of specific cell type transcriptomes from the neonatal (P7-17) mouse brain indicated significant levels of constitutive expression in microglial and endothelial cell populations, with considerably lower expression in other brain neuronal and glial cell types. (**D**) Specific ablation of *Tgfbr2* expression from retinal microglia of *Cx3Cr1 $^{CreER /+}$*,TGFBR2$^{flox/flox}$ (TG) mice was enabled by tamoxifen (TMX)-induced Cre recombinase activity in CX3CR1-expression microglia, resulting in the genetic excision of exon 4 of the *Tgfbr2* gene. CD11b+ microglia (*blue* points) and CD31+ endothelial cells (*green* points) were sorted from retinas of untreated TG mice, and from control and TG mice 3 weeks after tamoxifen administration using flow cytometry and analyzed. Cre recombinase-mediated excision of exon 4 of the *Tgfbr2* gene from the genomic DNA in microglial, endothelial, and the remaining retinal cell types (*purple* points) was assessed using qPCR; CD11b+ microglia demonstrated a significant
*Figure 1 continued on next page*

*Figure 1 continued*

loss of exon 4 relative to the exon 3 in TMX-treated TG animals, but not in untreated TG or TMX-treated control animals. Exon 4 excision was not observed in non-microglial cell types. (E) Quantitative rtPCR analysis demonstrated a corresponding reduction in the transcription of exon 4 of *Tgfbr2* mRNA from flow-sorted CD11b+ retinal microglia of TMX-treated TG animals relative to those of TMX-treated control animals and TG animals not treated with TMX. (Graphical data in (D) an (E) are presented as means ± SEM; p values are from one-way analysis of variance (ANOVA) and Sidak's multiple comparison test, n = 4 animals of mixed sex for each group).
DOI: https://doi.org/10.7554/eLife.42049.002

mRNA expression was high in both microglia and endothelial cells, with little or no expression in astrocytes, neurons, and oligodendrocytes (*Figure 1C*).

To evaluate the functional significance of constitutive in vivo TGFβ signaling in retinal microglia, we employed a transgenic mouse in which TGFBR2 can be specifically and inducibly ablated in CX3CR1-expressing microglia. We employed the $Cx3cr1^{CreER/+}$, $Tgfbr2^{flox/flox}$ (termed TG) mouse model in which tamoxifen administration activates $Cre^{ERT2}$ recombinase activity, enabling the excision of exon 4 of the *Tgfbr2* gene, ablating TGFβ signaling. Following induction, retinal CD11b+- microglia cells and CD31+ endothelial cells were isolated by flow sorting; experimental controls included age-matched TG animals that were not administered tamoxifen, and age-matched transgenic $Tgfbr2^{flox/flox}$ (termed Control) mice that lacked *Cre* recombinase and which were administered tamoxifen on the same dosing regimen as TG mice. Quantitative PCR analysis of genomic DNA for the targeted exon 4 of the *Tgfbr2* gene showed a significant reduction (relative to the preserved exon 3) in CD11b+ microglia isolated from tamoxifen-administered TG animals, but not in CD11b + microglia from both control groups (*Figure 1D*). No changes in the relative presence of exon 4 were detected in CD31+ endothelial cells or in the remaining (non-CD11b+, non-CD31+) retinal cell populations for all three experimental groups. Quantitative rtPCR analysis of mRNA isolated from flow-sorted CD11b+ retinal microglia correspondingly demonstrated a marked reduction of exon 4- containing transcripts in tamoxifen-administered TG animals but not in TG animals not administered tamoxifen, nor in tamoxifen-administered control mice (*Figure 1E*). These results indicate that *Tgfbr2* expression can be ablated in an inducible manner in adult TG animals and specifically in microglia among retinal cell types.

## Specific *in vivo* ablation of TGFBR2 in retinal microglia induces rapid morphological transformation and proliferation

We examined microglial morphology and distribution in the retinas of TG mice at different time points following tamoxifen-induced TGFBR2 ablation (1, 2, 5 days; 3 and 10 weeks) in flat-mounted samples. At one day post-tamoxifen administration, retinal microglia showed slight decreases in the length and branching of their processes but still retained a ramified morphology (*Figure 2*). At 2 to 5 days post-tamoxifen, microglia demonstrated marked reductions in process ramification, and possessed mostly short, stubbly processes. At 3 to 10 weeks post-tamoxifen, microglia transitioned to an elongated cellular morphology that had only a few processes that showed little branching. Interestingly, elongated microglia in TG animals were closely adherent to isolectin B4 (IB4)-labelled retinal blood vessels, with their processes conforming to the branched structure of retinal vessels (*Figure 2A,E*, *Videos 1* and *2*). These features were prominently seen at 3 weeks post-tamoxifen in both the OPL (*Figure 3A*) and the IPL (*Figure 3—figure supplement 1A*). Microglia in control animals administered tamoxifen were morphologically unchanged, resembling microglia in age-matched wild type animals not administered tamoxifen. Isolated Iba1+ microglia with elongated morphologies were also found in the subretinal space, a zone lacking retinal blood vessels (*Figure 3—figure supplement 1B*), indicating that the morphological transformations in microglia likely originated from cell-autonomous changes in microglia, rather than indirectly induced by signals from retinal vessels.

Morphological transformations in retinal microglia were accompanied by general increases in overall microglial density. Quantification of CD11b+ retinal microglia using flow cytometry showed significant increases in TG mice at 3 weeks and 10 weeks following tamoxifen administration compared with tamoxifen-administered control mice (*Figure 3B*). Microglia densities, as assessed by cell counting in flat-mounted retinal specimens using immunochemical analyses, also demonstrated increases in all retinal laminae (IPL, OPL, and SRS) (*Figure 3C*), which corresponded to the

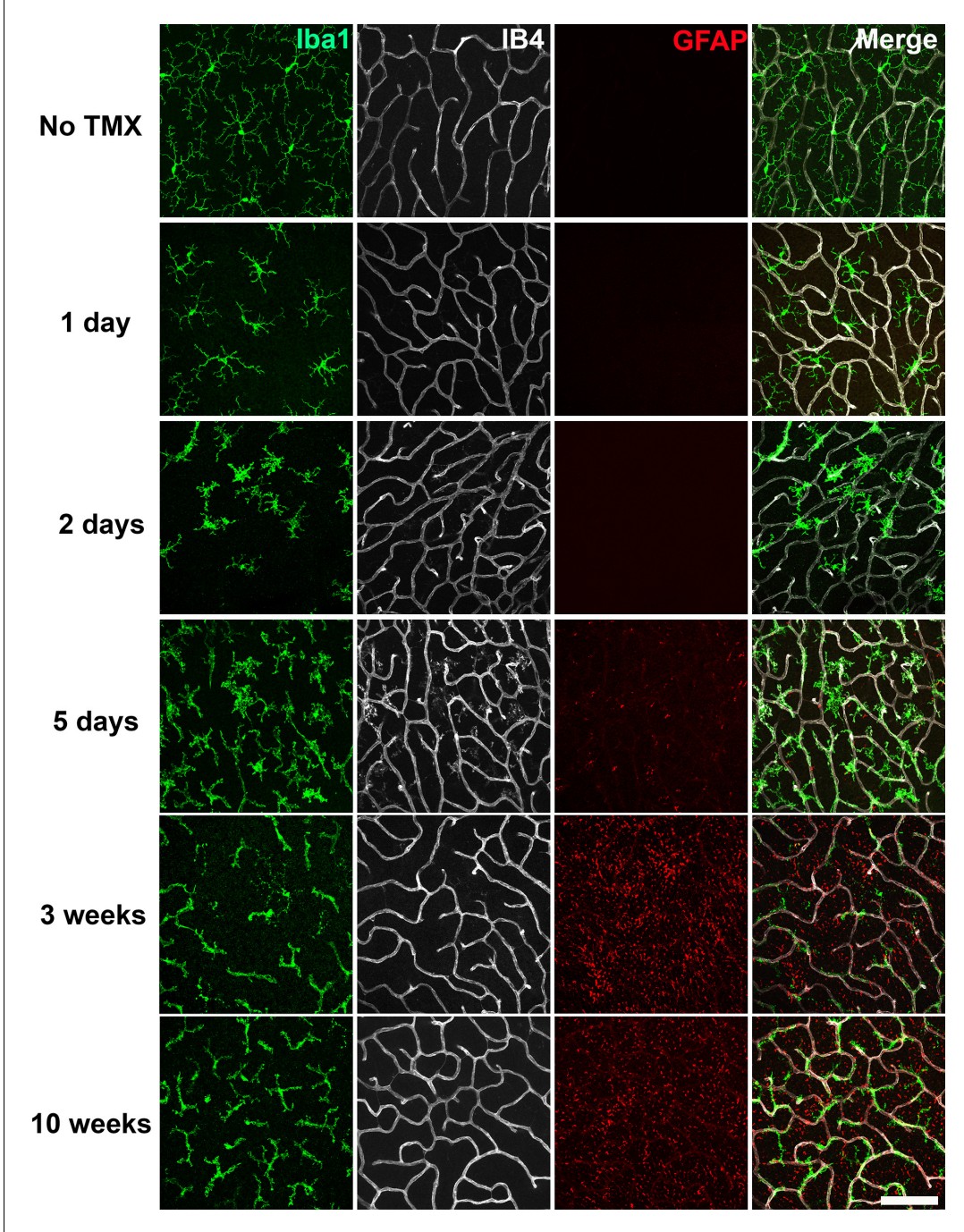

**Figure 2.** Specific TGFBR2 ablation in retinal microglia induces rapid and progressive changes in microglial morphology and distribution. The time course of morphological changes in retinal microglia following tamoxifen (TMX)-induced ablation of TGFBR2 expression was followed using immunohistochemical analysis in retinal flat-mounts. Panels show changes at the level of the OPL; microglia were labeled using an antibody to IBA1 and retinal vessels labeled with IB4. Gliotic changes in radial Müller glia processes were marked using an antibody to GFAP. At 1 day following TMX administration, a slight reduction in ramification in microglia processes was observed. From 2–5 days post-TMX, a further decrease in microglial ramification and an increase in microglia numbers were detected. From 3–10 weeks post-TMX, retinal microglia transitioned to a branched morphology, demonstrating a close fasciculation with the retinal vasculature. GFAP immunopositivity in Müller glia was prominently upregulated at this time. Scale bar = 100 μm.

DOI: https://doi.org/10.7554/eLife.42049.003

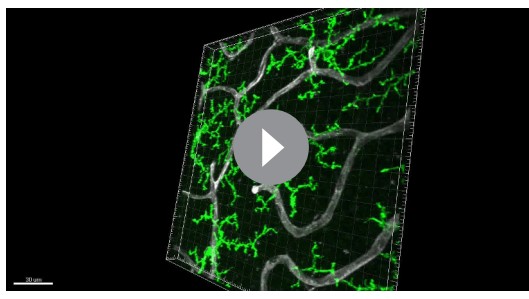

**Video 1.** 3D rotation depiction of the morphology and distribution of IBA1-immunolabelled retinal microglia (*green*) in the OPL with respect to IB4-labeled retinal vessels (*white*) in TG animals prior to the administration of tamoxifen.
DOI: https://doi.org/10.7554/eLife.42049.007

emergence of proliferating, Ki67+, microglia. (*Figure 3D*). Ki67 immunopositivity was absent in subretinal microglia, suggesting that increased subretinal microglia numbers may have resulted from migration of microglia from the inner retinal layers. Monocytic infiltration into the retina was unlikely to have contributed to the increased IBA + cell numbers as cell-fate mapping of retinal microglia vs. systemic monocytes using TG mice crossed into the Ai14 background (*Ma et al., 2017*) revealed that IBA1+ cells 4 months following tamoxifen uniformly expressed tdTomato, indicating that systemic monocytes (which at this time had been turned over and replaced by tdTomato-negative cells) had not contributed to the increased numbers of IBA1+ cells induced by TGFBR2 ablation (*Figure 3—figure supplement 1C*). In addition, we observed that the myeloid cells in the retina demonstrating progressive mor-

phological change in the first week following tamoxifen administration were immunopositive for P2RY12, a marker for endogenous microglia, as well as for Ki67, a marker of proliferating cells (*Figure 3—figure supplement 1D*). Although P2RY12 immunopositivity was gradually lost after one week following TGFBR2 ablation, these findings indicated that the population of morphologically-transformed myeloid cells in the retina arose from the proliferation and modification of pre-existing endogenous retinal microglia.

As the constitutive presence of microglia in the adult retina is required for ongoing maintenance of retinal synapses (*Wang et al., 2016*), and may be mediated by repeated microglia-synapse contacts via dynamically motile microglial processes (*Lee et al., 2008*), we examined how areal coverage in the synapse-rich plexiform layers by microglial processes may be altered following TGFBR2 ablation. We found that following TGFBR2 ablation, individual microglial cells in both the IPL and OPL of TG animals demonstrated marked reductions in the number of branch points per cell (*Figure 3E,F*) and in the area subtended by the processes of each cell (*Figure 3G*). Consequently, despite increased microglial density, the proportion of retina lacking direct coverage by microglial processes was significantly greater (*Figure 3H*), translating to decreased microglia-synapse contact and likely diminished microglial supportive functions.

## Specific ablation of TGFBR2 in retinal microglia results in decreased microglial 'sensome' function and increased activation

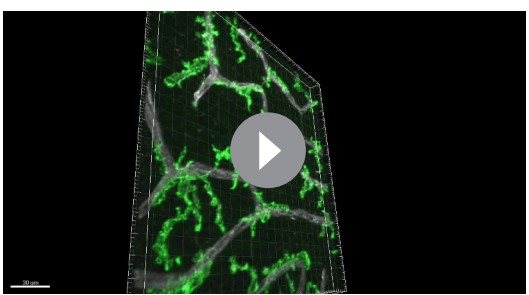

**Video 2.** 3D rotation depiction of the morphology and distribution of IBA1-immunolabelled retinal microglia (*green*) in the OPL with respect to IB4-labeled retinal vessels (*white*) in TG animals 2 weeks following the administration of tamoxifen to induce TGFBR2 ablation in retinal microglia.
DOI: https://doi.org/10.7554/eLife.42049.008

Corresponding to the decreased spatial coverage of the retina by TGFBR2-ablated microglia, we investigated if the ability of retinal microglia to sense environmental signals may be affected by the loss of TGFβ signaling. Previous transcriptomic profiling studies of microglia in the mouse brain have defined a cluster of microglial specific/enriched transcripts encoding proteins that confer the ability to sense environmental signals, collectively referred to as the microglial 'sensome' (*Hickman et al., 2013*). Quantitative RT-PCR analysis of flow-sorted microglia from the TG retinas 2 weeks following tamoxifen administration revealed that mRNA expression levels of 'sensome' transcripts such as *Cx3cr1*, *P2yr12*, *Tmem119,* and *Siglech*, were markedly reduced relative to control mice (*Figure 4A*). These

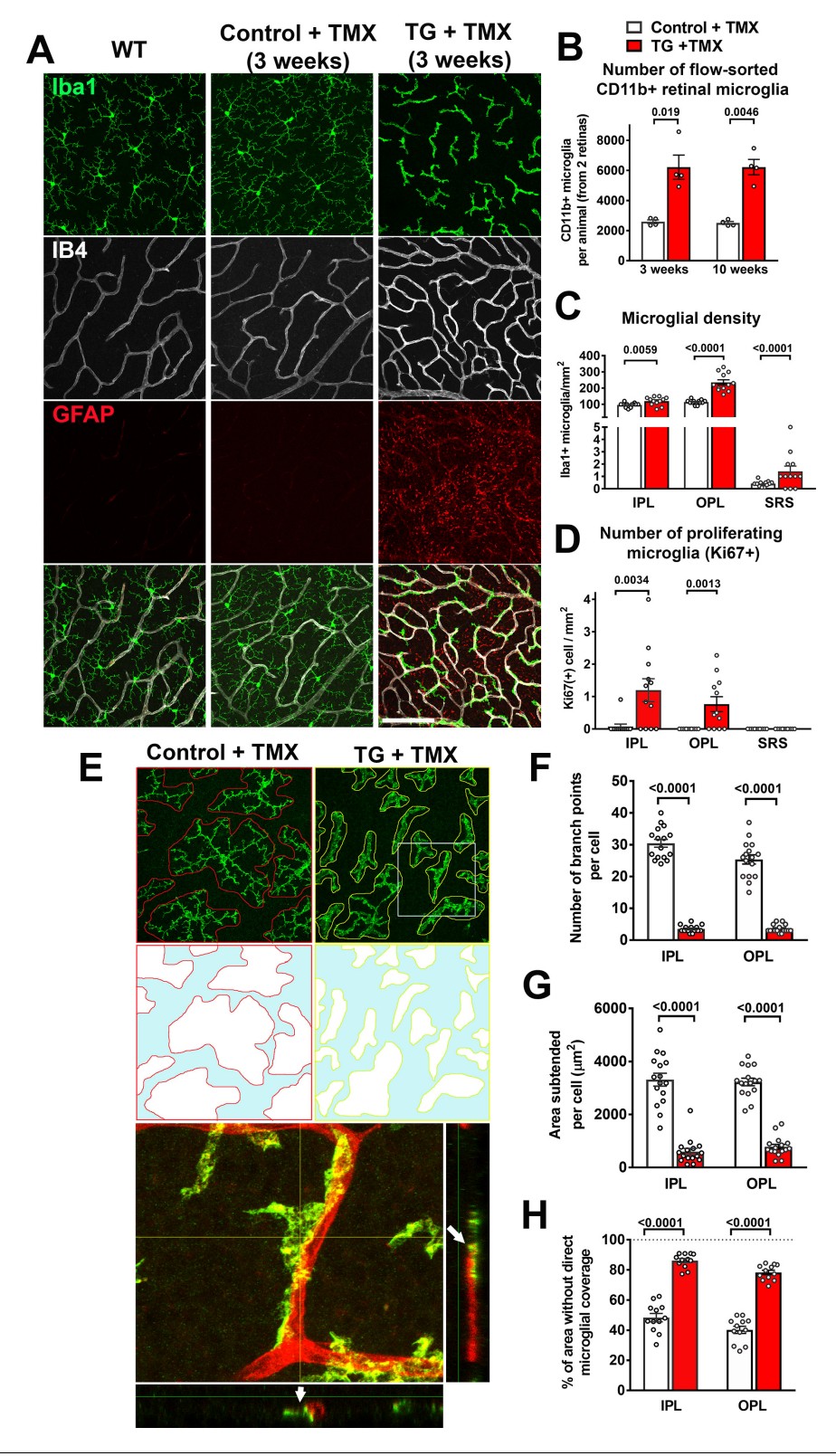

**Figure 3.** Specific TGFBR2 ablation in retinal microglia induces abnormalities in microglial density, distribution, and morphology. (**A**) TG animals administered tamoxifen (TMX) 3 weeks prior, relative to wild type (WT) mice and control mice, demonstrated that TGFBR2 ablation resulted in increased microglial numbers and decreased ramification in the OPL. Scale bar = 100 μm. (**B**) Analysis of CD11b+ microglia numbers in each animal (two retinas combined) using flow-cytometry showed a significant increase in microglial numbers in TG vs. control animals at 3- and 10 weeks post-TMX.
*Figure 3 continued on next page*

*Figure 3 continued*

Manual counts of Iba1 +microglia numbers (**C**) and proliferating Ki67+, Iba1 +microglia (**D**) in retinal flat-mounts from animals 4 weeks post-TMX demonstrated increases at the levels of the IPL, OPL, and subretinal space (SRS) in TG vs. control retina. (**E**) TGFBR2-ablated microglia 4 weeks post-TMX showed reduced process ramification and decreased dendritic area (as highlighted in outlines of individual microglial dendritic arbors). TGFBR2-ablated microglia demonstrated branched morphologies (example shown in yellow box, expanded in inset) that showed close adherent contact with IB4-labelled (*red*) retinal vessels (*arrows* indicating points of contact). Scale bar = 100 μm. Morphological analysis of individual microglia showed significant decreases in the number of branch points (**F**) and in the areas of individual arbors (**G**) of microglia in both IPL and OPL in TGFBR2-ablated microglia. Despite having increased numbers of total microglia, TMX-treated TG retinas have a greater proportion of retinal area not directly occupied by microglial processes (**H**, areas highlighted in *blue*), indicating decreased microglial coverage. Graphical data are presented as means ± SEM; p values are from unpaired t-test with Welch's correction, data points in (**C**), (**D**), and (**H**) represent four individual imaging fields from three animals in each group, those in (**F**) and (**G**) represent 16 individual microglia cells from four animals in each group).

DOI: https://doi.org/10.7554/eLife.42049.004

The following figure supplements are available for figure 3:

**Figure supplement 1.** Specific TGFBR2 ablation in retinal microglia induces alterations in microglial morphology in the IPL and SRS.

DOI: https://doi.org/10.7554/eLife.42049.005

**Figure supplement 2.** TGFBR2 ablation in retinal microglia induces expression of CD206, a marker associated with perivascular macrophages.

DOI: https://doi.org/10.7554/eLife.42049.006

changes are likely to be a direct consequence of the loss of TGFβ signaling in retinal microglia as *in vitro* administration of TGFβ ligands (TGFB1, TGFB2) to cultured retinal microglia isolated from wild type mice resulted in upregulation of sensome transcripts, *Tmem119* and *Siglech* (*Figure 4B*). Down-regulation of sensome gene expression with TGFBR2-ablation was also apparent on a protein level; fluorescence associated with EYFP expression as driven by the *Cx3cr1* promoter in TG mice, a surrogate marker for the level of *Cx3cr1* expression, was significantly reduced by TGFBR2 ablation (*Figure 4C,D*), which was also associated with decreased TMEM119 immunopositivity in Iba1+ microglia (*Figure 4E,F*). Microglial responses to endogenous signals include the provision of trophic support to nearby neurons in the form of growth factors, such as BDNF (*Parkhurst et al., 2013*) and IGF1 (*Lalancette-Hébert et al., 2007*). We found that mRNA expression of growth factors, *Bdnf* and *Pdgfa*, were decreased in retinal microglia following TGFBR2 ablation, while that for *Igf1* was unchanged (*Figure 4—figure supplement 1*). Accordingly, the addition of TGFβ ligands also increased the expression of these growth factors in isolated retinal microglia in culture. Together, these observations indicated that TGFβ-signaling to microglia sustains the microglial homeostatic gene signature and promotes the ability of microglia to sense endogenous signals and exert trophic influences in the retina.

As TGFβ signaling has been associated with the induction of a quiescent microglial phenotype in the brain (*Abutbul et al., 2012*), we evaluated if genetic ablation in microglia within the retina influenced their activation status. We found that TGFBR2 ablation in microglia upregulated mRNA expression of activation markers (MHCII (*H2-Aa*), *Cd68*, *Cd74*), chemotactic cytokines (*Ccl2* and *Ccl8*), and *Apoe*, a promoter of a proinflammatory, disease-associated microglial phenotype (*Kang et al., 2018*; *Krasemann et al., 2017*) (*Figure 5A*). Cultured WT retinal microglia demonstrated corresponding decreases in *ApoE* and *Ccl2* mRNA levels when TGFβ ligands were added in vitro (*Figure 5B*). Immunohistochemical analysis of TG mice following tamoxifen administration showed increased immunopositivity for markers of microglial activation, including CD68, MHCII (*Figure 5C–F*), CD74, F4/80, and CD45 (*Figure 5—figure supplement 1A–F*) relative to control mice. RT-PCR analysis of mRNA expression in the retina following microglial TGFBR2 ablation also found progressively increasing expression of transcripts found to be enriched in macrophages over that in homeostatic microglia (*Saa3, Pf4, Cd5l*) (*Hickman et al., 2013*) (*Figure 5—figure supplement 1G*). These data indicated that constitutive direct TGFβ signaling is required for the general suppression of microglial activation.

## TGFBR2-deficient microglia induce secondary Müller cell gliosis and neuronal degeneration in the surrounding retina

We examined the consequences of microglia-specific TGFBR2 ablation to the structure and function of the surrounding retina. Following tamoxifen administration in TG mice, we observed an emergence of a radial pattern of GFAP immunopositivity beginning at 5 days which persisted at 10 weeks

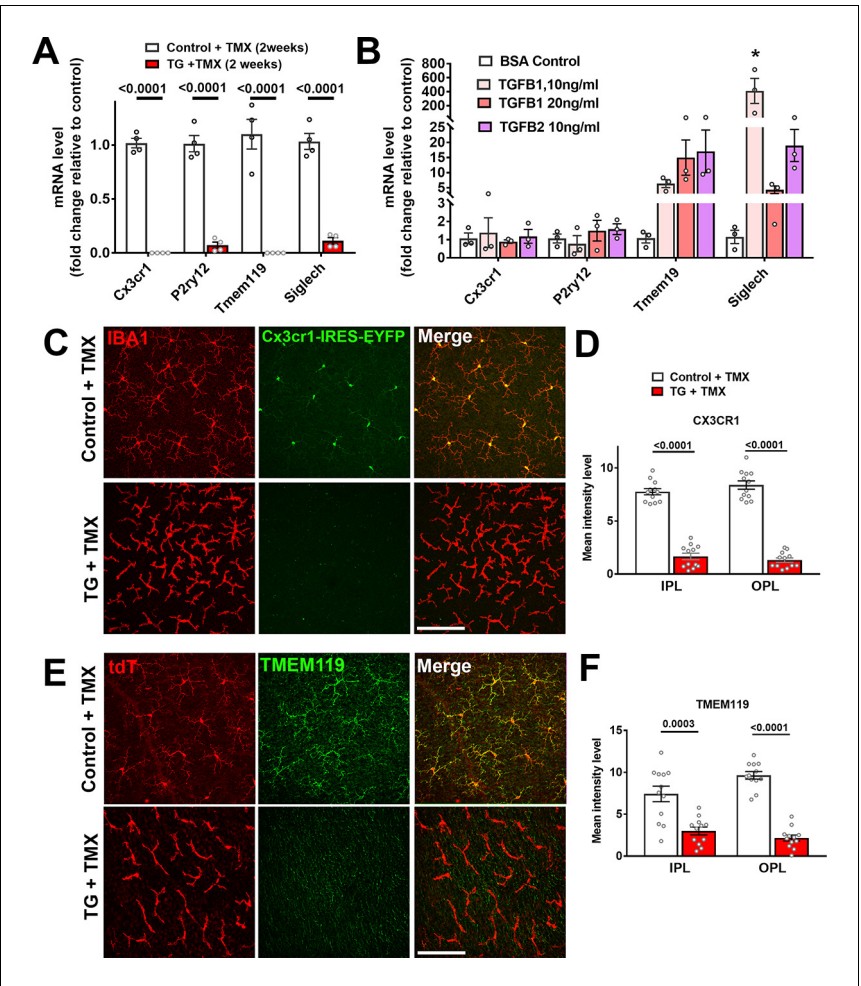

**Figure 4.** Constitutive expression of microglial 'sensome' genes are downregulated upon TGFBR2 ablation in retinal microglia. (A) Retinal microglia from control and TG mice were isolated by flow-cytometry 2 weeks following tamoxifen (TMX) administration and mRNA levels of microglial 'sensome' genes compared using qPCR. mRNA levels of *Cx3cr1*, *P2yr12*, *Tmem119*, and *Siglech* were all significantly decreased in microglia from TG vs. control mice. (B) Microglia from the retinas of WT mice were cultured and exposed to media containing TGFB1 (10 or 20 ng/ml), or TGFB2 (10 ng/ml) (media containing 10 ng/ml of BSA served as a control), and mRNA levels of microglial 'sensome' genes compared following 24 hr of exposure. mRNA levels of *Tmem119* and *Siglech* were increased by TGFBR2 ligands (TGFB1 or TGFB2), indicating positive regulation of microglial 'sensome' genes via TGFBR2-mediated signaling. (C, D) As TG animals contained an IRES-EYFP cassette 3' to CreERT recombinase in the *Cx3cr1* locus, EYFP expression, as regulated by the *Cx3cr1* promoter, could be constitutively detected in IBA1-immunopositive retinal microglia in control animals. In TG animals at 3 weeks post-TMX, *Cx3cr1*-driven EYFP fluorescence was diminished in Iba1+ microglia, indicating downregulation of *Cx3cr1* promoter activity. (E, G) Immunohistochemical analysis of TMEM119 showed strong colocalization with Iba1 in microglia of control animals but decreased immunopositivity in TGFBR2-ablated microglia in TG animals. Scale bars = 100 μm. Graphical data in (A), (B), (D) and (F) are presented as means ± SEM; p values in (A), (D), and (F) are from multiple t-tests, while that in (B) are from 2-way ANOVA analysis with Sidak's multiple comparisons test, * indicate p<0.05 for comparisons relative to control, data points indicate individual biological repeats in (A) and (B), and four imaging fields from three animals in each group in (D) and (F).

DOI: https://doi.org/10.7554/eLife.42049.009

The following figure supplement is available for figure 4:

**Figure supplement 1.** Expression of growth factors genes are downregulated in microglia upon TGFBR2 ablation.
DOI: https://doi.org/10.7554/eLife.42049.010

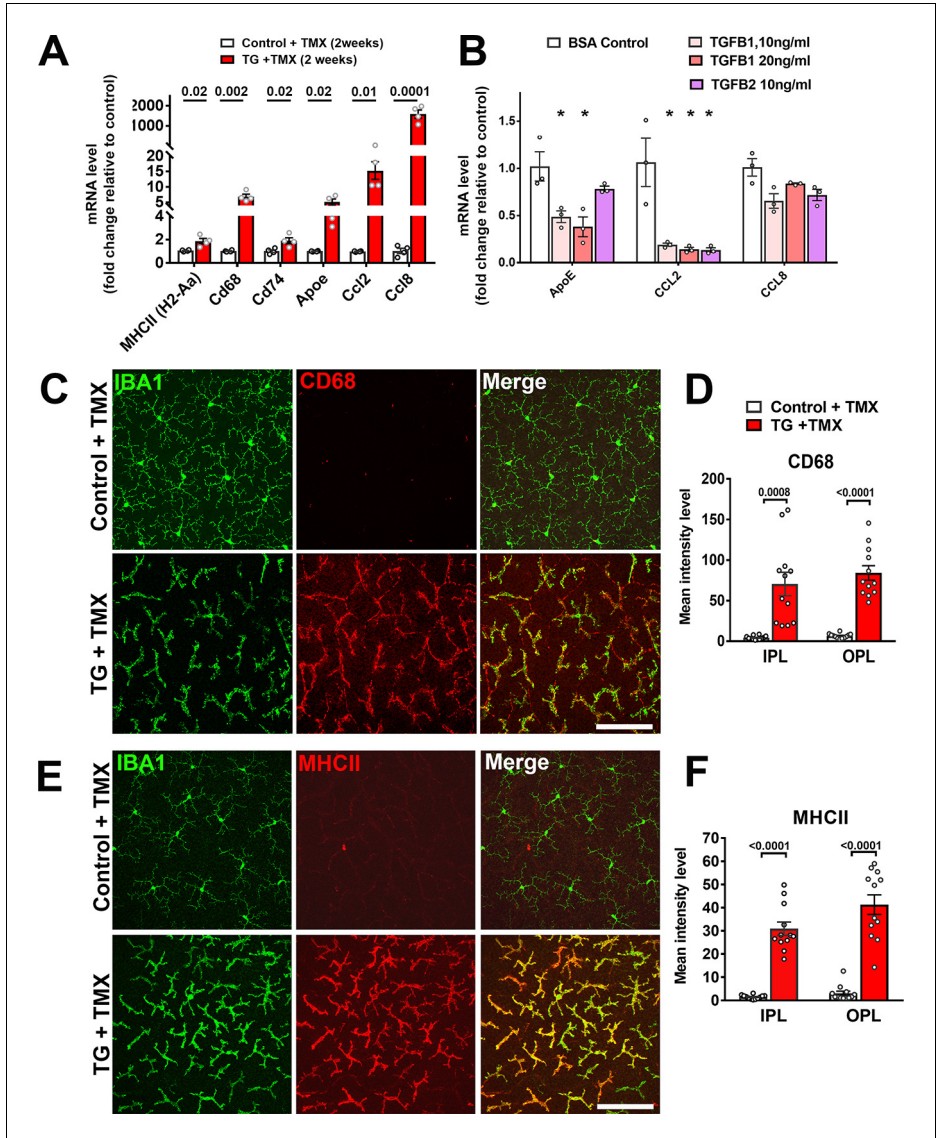

**Figure 5.** Expression of genes associated with microglial activation are upregulated on TGFBR2 ablation in retinal microglia. (**A**) Retinal microglia from control and TG mice were isolated by flow-cytometry 2 weeks following tamoxifen (TMX) administration and mRNA levels of genes associated with microglial activation and inflammatory chemokines were analyzed and compared using qPCR. mRNA levels for *H2-Aa* (MHCII), *Cd68, Cd74, Apoe, Ccl2,* and *CCl8* were all significantly increased in microglia from TG vs. control mice. (**B**) Microglia from the retinas of WT mice were cultured and exposed to media containing TGFB1 (10 or 20 ng/ml), or TGFB2 (10 ng/ml) (media containing 10 ng/ml of BSA served as a control), and mRNA levels of microglial-expressed genes compared following 24 hr of exposure. mRNA levels of *Apoe* and *Ccl2* were decreased by TGFBR2 ligands (TGFB1 or TGFB2), indicating negative regulation of microglial activation genes via TGFBR2-mediated signaling. Immunohistochemical analysis of control vs. TG microglia in retinal flat-mounts showed prominent and significant upregulation of activation markers CD68 (**C, D**) and MHCII (**E, F**) in Iba1+ microglia in both the IPL and OPL. Scale bars = 100 µm. (Graphical data in (**A**), (**B**), (**D**) and (**F**) are presented as means ± SEM; p values in (**A**), (**D**), and (**F**) are from multiple t-tests, while that in (**B**) are from 2-way ANOVA analysis with Sidak's multiple comparisons test, * indicate p<0.05 for comparisons to control, data points indicate individual biological repeats in (**A**) and (**B**), and four imaging fields from 3 to 4 animals in each group in (**D**) and (**F**)).

DOI: https://doi.org/10.7554/eLife.42049.011

The following figure supplement is available for figure 5:

**Figure supplement 1.** Specific TGFBR2 ablation in microglia induces expression of markers of microglial activation.

*Figure 5 continued on next page*

*Figure 5 continued*

DOI: https://doi.org/10.7554/eLife.42049.012

(*Figure 2*) that colocalized with glutamine synthetase (GS)-immunopositive Müller cells processes (*Figure 6A*). GFAP mRNA levels in the retina were also increased at 2 weeks following TGFBR2 ablation and persistent at 8 weeks (*Figure 6B*). Also, a progressive upregulation of mRNA levels for genes associated with neurotoxic A1 astrocytic gliosis (*Liddelow et al., 2017*) was induced, while those for genes associated with the neuroprotective form of A2 gliosis were relatively unchanged

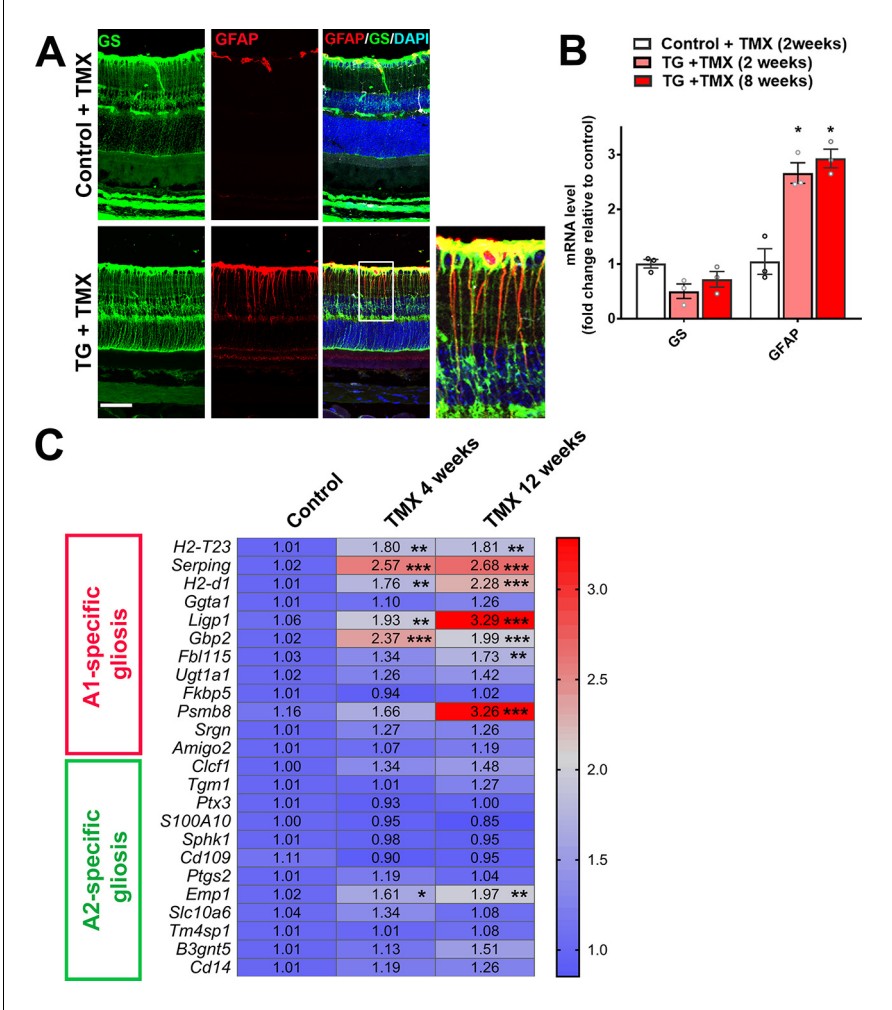

**Figure 6.** TGFBR2 ablation in retinal microglia induces Müller cell gliosis in the retina. (**A**) Immunohistochemical analysis demonstrates upregulation of immunopositivity to GFAP 3 weeks post-TMX in TG animals relative to control animals. GFAP immunopositivity was localized to glutamine synthetase (GS)-labeled Müller cell processes, indicating the induction of Müller cell gliosis. Scale bar = 50 μm. (**B**) qPCR analysis of retinas isolated from control and TG animals 2 and 8 weeks post-TMX demonstrates a significant upregulation of GFAP mRNA expression following TGFBR2 ablation in retinal microglia. Graphical data are presented as means ± SEM; p values are from one-way analysis of variance (ANOVA) and Sidak's multiple comparison test, n = 3 animals of mixed sex in each group.(**C**) RT-PCR analysis of retinal expression of genes associated with A1- and A2-specific astrocytic gliosis following microglial TGFBR2 ablation found progressive upregulation of A1-associated transcripts relative to control, while A2-associated transcripts were relatively unchanged (numbers indicate means, \*, \*\*, \*\*\* indicate p values < 0.05,<0.01,<0.001 respectively, 2-way ANOVA analysis with Sidak's multiple comparisons test, data from 3 to 4 animals in each group.).

DOI: https://doi.org/10.7554/eLife.42049.013

(*Figure 6C*). These observations indicate that microglial transformation induced by TGFBR2 ablation led to a rapid and durable induction of gliotic changes in surrounding Müller cells that resemble reactive A1 astrocytic gliosis characterized in the brain under conditions of neurodegeneration and aging (*Clarke et al., 2018*; *Liddelow et al., 2017*).

As chronic proinflammatory microglial activation in the retina has been associated with neuronal degeneration (*Langmann, 2007*), we investigated if microglial alterations following TGFBR2 ablation resulted in deleterious changes in retinal neurons. Using *in vivo* optical coherence tomography (OCT) imaging we found total retinal thickness in TG mice decreased progressively with time following tamoxifen administration, falling to 95% of controls at 3 weeks and to 80% at 10 weeks (*Figure 7A,B*). These changes were contributed to by decreases in retinal thickness in the inner, as well as in the outer retina. Quantitative assessment of the thickness of retinal laminae from histological retinal sections also showed significant decreases in inner and outer nuclear layers, as well as the inner and outer plexiform layers (*Figure 7C,D*). Analysis in flat-mounted retinal samples demonstrated significant decreases in the density of BRN3A-immunopositive retinal ganglion cells and cone arrestin-positive cone photoreceptors (*Figure 7—figure supplement 1*). These changes were correlated with the appearance of apoptotic TUNEL +nuclei in all retinal nuclear layers, indicating an induction of neuronal apoptosis (*Figure 7E,F*). Assessment of retinal function using electroretinography (ERG) revealed that the amplitudes of dark-adapted, rod photoreceptor-dominant responses were significantly reduced in TG vs. control animals, with b-wave amplitudes significantly more reduced than a-wave amplitudes (*Figure 7G*). For light-adapted, cone photoreceptor-mediated responses, only b-wave amplitudes were significantly reduced, while a-wave amplitudes were unchanged (*Figure 7H*). Overall, significant decreases in the b-to-a amplitude ratios were observed for both dark- and light-adapted responses, indicating that ablation of TGFBR2 in retinal microglia resulted in some measure of rod photoreceptor dysfunction and also a loss of synaptic transmission in both rod and cone photoreceptors, as previously described for the ablation of microglia in the adult retina (*Wang et al., 2016*). Taken together, the physiological switch of retinal microglia from a homeostatic mode to a more activated mode upon the loss of microglia TGFβ signaling, is likely causally associated with a loss of microglial support, a dysregulation of inflammatory responses resulting in gliotic changes, and the induction of neuronal and synaptic degeneration.

## Molecular pathways underlying retinal changes induced by microglial TGFBR2 ablation

To further investigate the nature of molecular pathways underlying retinal changes following microglial TGFBR2 ablation, we profiled the mRNA expression levels of 547 immunology-associated genes using targeted multiplex analysis (nCounter, Nanostring). We compared expression in retinas isolated from 4 groups of animals: two groups of control animals with and without tamoxifen administration, and two groups of TG animals administered tamoxifen for either 2 or 8 weeks. Hierarchical clustering revealed similar mRNA profiles between control animals with or without tamoxifen administration, indicating that tamoxifen *per se* did not exert a major effect on inflammatory retinal gene expression (*Figure 8A*), while tamoxifen-administered TG animals showed significant differences from control animals. Significantly upregulated genes (>2 fold increase in expression, p<0.05) following either 2 or 8 weeks of tamoxifen administration in TG animals included: (1) markers of microglial activation, such as *Cd74*, *H2-Aa* (MHCII), *Cd163*, C*d40*, (2) proinflammatory cytokines, such as *Ccl2*, *Ccl4*, *Ccl12*, and *Ccl8*, (3) complement components, such as *C4a* and *C3*, (4) regulator of inflammatory responses, such as FcγR receptors and *Casp*1 (*Figure 8B*). Early downregulation of the microglial sensome gene *Cx3cr1* was also detected at 2 weeks. Gene ontology (GO) analysis of differentially expressed genes revealed the differential involvement of pathways in neuroinflammatory signaling and nuclear factor of activated T-cells (NFAT)-mediated signaling, which has been implicated in the regulation of microglial activation (*Nagamoto-Combs and Combs, 2010*) (*Figure 8C*). Network analysis indicated that the differentially expressed genes associated with microglial TGFBR2 ablation related to various aspects of cell function including (1) homeostasis of leukocytes, (2) inflammatory response, (3) cytotoxicity, and (4) activation, which may be potentially regulated by interferon-α, interferon-γ, IL1β signaling, and NFKB-regulated transcription (*Figure 8—figure supplement 1*).

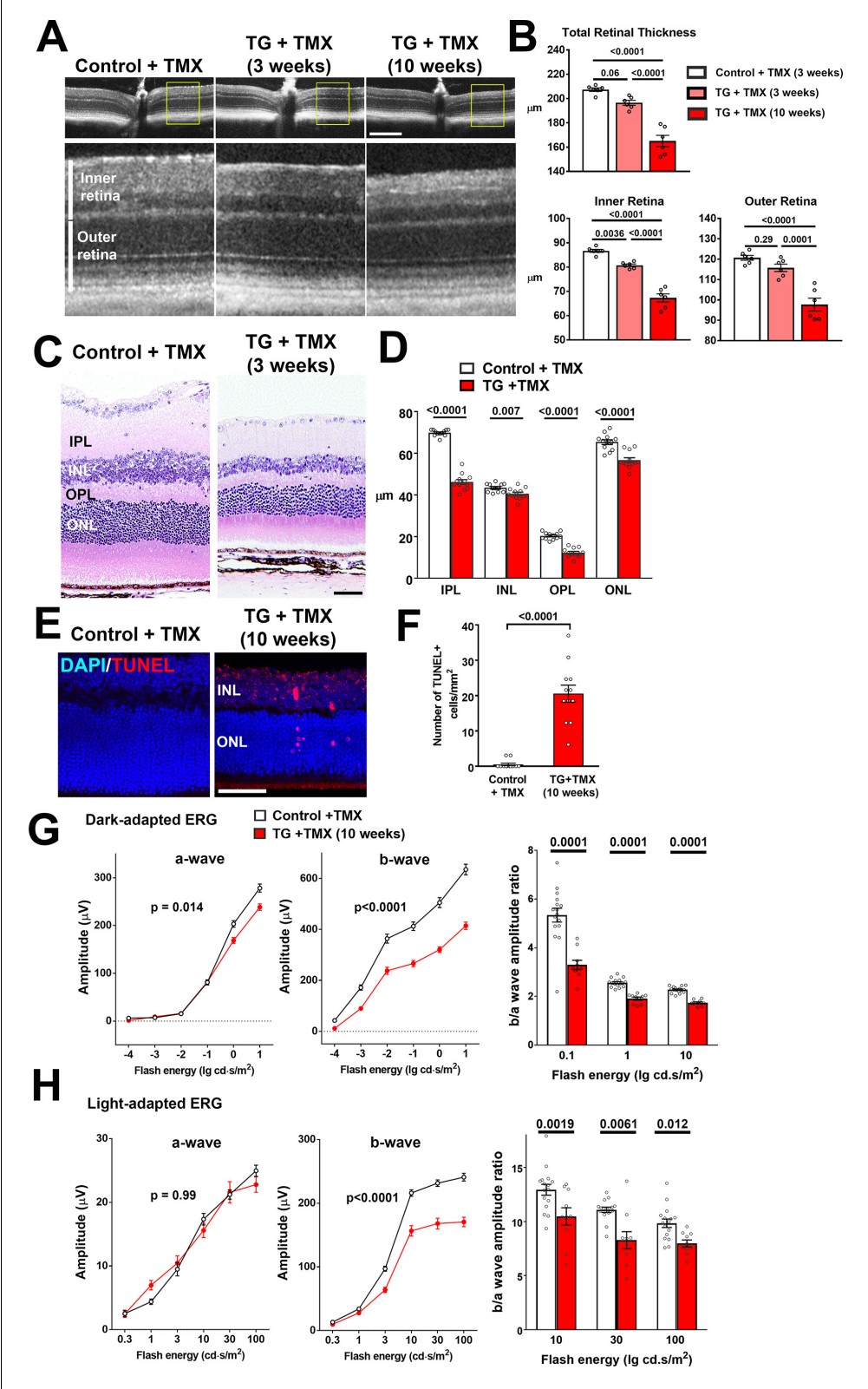

**Figure 7.** TGFBR2 ablation in retinal microglia induces degenerative changes in the retina. (**A, B**) *In vivo* evaluation of retinal structure by optical coherence tomography (OCT) in control animals and in TG animals 3 and 10 weeks following tamoxifen (TMX)-administration showed a preserved lamination in TG animals (insets at higher magnification in *yellow boxes*) but a progressive and significant reduction in the total retinal thickness relative to controls. Scale bar = 300 μm. Significant reductions in overall thickness were contributed to by reductions in both the inner (measured from vitreal

*Figure 7 continued on next page*

*Figure 7 continued*

surface to the outer plexiform layer) and the outer retinal layers (measured from the outer plexiform layer to the apical surface of the RPE layer) (p values are from 1-way ANOVA analysis with Tukey's multiple comparisons test, data points are from 6 eyes of 3 animals). (C, D) Histological analysis of retinal lamina thicknesses in paraffin-embedded sections show significant decreases in the thickness of the inner plexiform layer (IPL), inner nuclear layer (ONL), outer plexiform layer (OPL), and outer nuclear layer (ONL) in TG animals 3 weeks post-TMX relative to controls (p values are from unpaired t-tests with Welch's correction, data points are from 3 sections from four animals). Scale bar = 50 µm. (E, F) Evaluation for apoptotic retinal cells using TUNEL labeling demonstrated the emergence of apoptotic cells in both the INL and ONL in TG retinas 10 weeks post-TMX. (p values are from unpaired t-tests with Welch's correction, data points are from 3 sections from four animals). Scale bar = 50 µm. (G, H) Comparison of electroretinographic (ERG) responses between control vs. TG animals 10 weeks post-TMX demonstrated in dark-adapted responses (G) a small but significant decrease in a-wave amplitude and a marked decrease in b-wave amplitudes in TG animals. Light-adapted responses (H) were similar for a-wave amplitude but significantly decreased in b-wave amplitude. The b-to-a amplitude ratios were significantly decreased in TG animals in both dark- and light-adapted responses for a range of flash intensities (p values are from 2-way ANOVA analysis, data points are both eyes of 8 control and 8 TG animals).

DOI: https://doi.org/10.7554/eLife.42049.014

The following figure supplement is available for figure 7:

**Figure supplement 1.** TGFBR2 ablation in retinal microglia induces degenerative loss of retinal neurons.

DOI: https://doi.org/10.7554/eLife.42049.015

## TGBR2 ablation in microglia does not affect the normal structure of retinal blood vessels but promotes pathological choroidal neovascularization

As TGFBR2 ablation in the entire retina at an early age (*Braunger et al., 2015*) or specifically in retinal endothelial cells (*Allinson et al., 2012*; *Schlecht et al., 2017*) resulted in abnormal development and structure of retinal blood vessels, we investigated whether TGFBR2 ablation specifically in adult retinal microglia may result in similar effects. We found that 12 weeks following tamoxifen administration in TG mice, despite increases in the expression of inflammatory genes in the retina and the close adherence of deramified microglia to retinal vessels, the blood-retina barrier on fluorescein angiography remained relatively intact, with no obvious vascular leakage or changes in overall retinal perfusion (*Figure 9A*). In the absence of additional injury, microglial TGFBR2 ablation did not result in histological abnormalities in retinal vascular structure; the arrangement of retinal endothelial cells (marked by CD31 labeling and IB4 staining) and pericytes (marked by NG2 labeling) were similar to that in controls (*Figure 9B*), and lacked signs of pericyte and endothelial cell loss with spontaneous neovascularization previously described for TGFBR2 ablation in retinal endothelial cells (*Allinson et al., 2012*; *Schlecht et al., 2017*). However, when we induced choroidal neovascularization (CNV) using a laser injury model (*Campos et al., 2006*) and compared the neovascular pathology in tamoxifen-administered control and TG mice, we observed in TG mice an increased number of Iba1+ microglia/macrophages recruited to the site of laser injury, which was associated with a larger RPE layer defect and an increased size of the CNV complex (*Figure 9C-F*) . This indicated that TGFBR2 loss in microglia, while not inducing vascular change on its own, increased microglia recruitment to retinal injury and promoted CNV growth at the site of microglial aggregation. This data indicates that while microglial TGFBR2 expression was dispensable for the maintenance of normal retinal vasculature, microglia lacking TGFβ signaling can transition to phenotypes that can potentiate pathological neovascularization in the presence of inducing factors.

## Discussion

TGFβ signaling exerts pleiotropic effects in various tissues that mediate a broad range of regulatory influences on cell survival and inflammation (*Fabregat et al., 2014*; *Travis and Sheppard, 2014*). In the retina, TGFβ signaling is constitutively operational under healthy conditions, regulating the maintenance of normal retinal structure and function. TGFβ ligands, TGFβ1, TGFβ2, and TGFβ3, are expressed by multiple retinal cell types, including different classes of retinal neurons, endothelial cells, RPE cells, and retinal microglia (*Anderson et al., 1995*; *Close et al., 2005*; *Lutty et al., 1993*). In particular, *Tgfb2* and *Tgfb3* mRNA have been detected in amacrine, bipolar, and retinal ganglion cells (*Siegert et al., 2012*), and TGFB2 protein has been localized to photoreceptors (*Lutty et al., 1991*). TGFβ receptors are also broadly expressed in different retinal cell types (*Obata et al., 1999*);

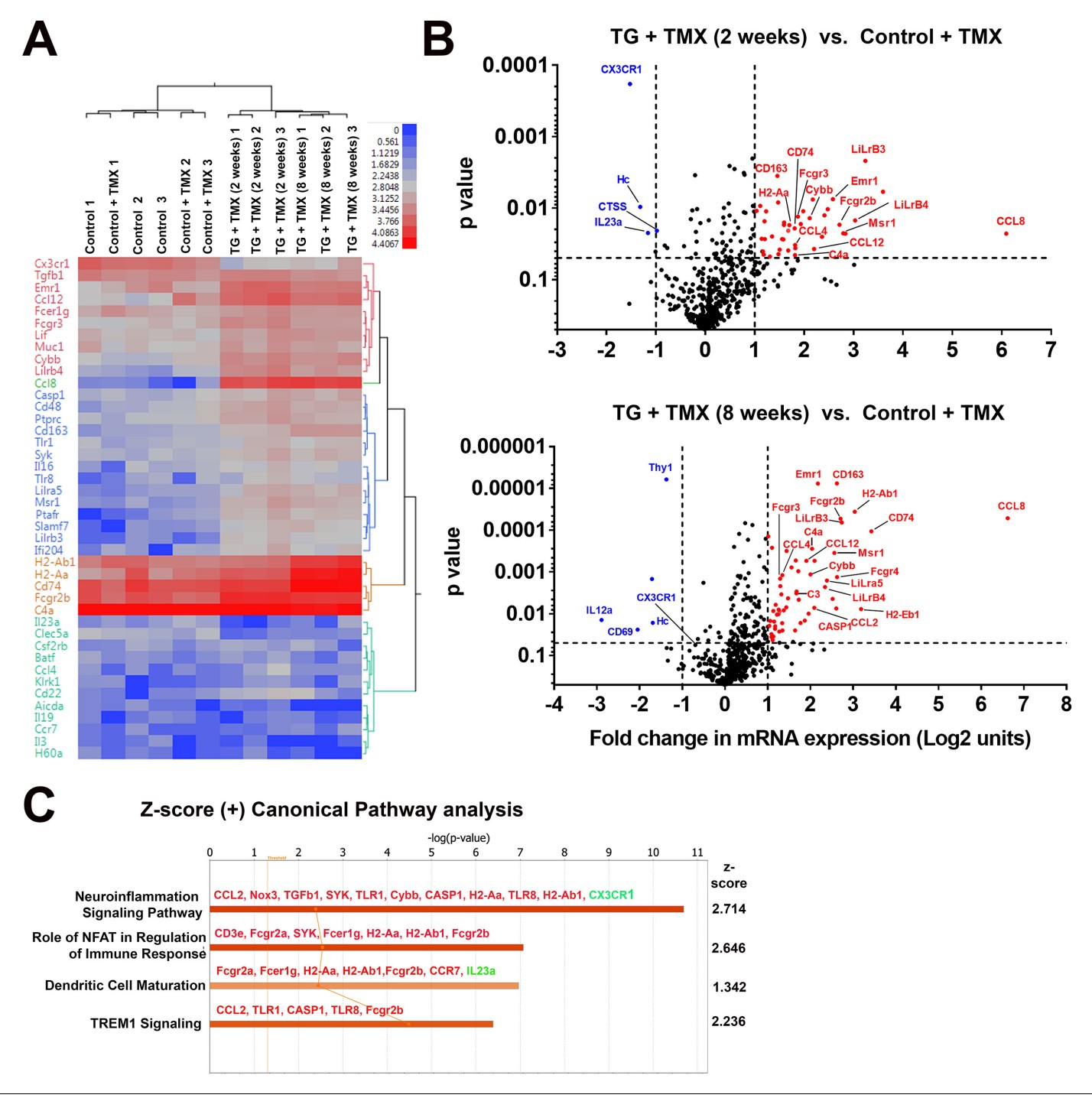

**Figure 8.** Changes in the mRNA expression of immune regulated genes in the retina following microglial TGFBR2 ablation using Nanostring-based profiling. Four groups of animals (n = 3 animals per group) were analyzed: (1) Control animals not administered tamoxifen, (2) Control animals administered tamoxifen, (3) TG animals 2 weeks after tamoxifen administration, (4) TG animals 8 weeks after tamoxifen administration. (A) Hierarchical clustering of differentially expressed genes showed separate clustering of control and TG animals administered tamoxifen. (B) Volcano plots showing genes that were differentially expressed between control and TG animals administered tamoxifen at 2 and 8 weeks respectively. (C) Gene ontogeny (GO) analysis using IPA demonstrated a number of canonical pathways that were differentially represented between control and TG animals administered tamoxifen at 2 weeks reflecting the activation of neuroinflammatory pathways and pathways involved in immune cell activation and maturation.

DOI: https://doi.org/10.7554/eLife.42049.016

*Figure 8 continued*

The following figure supplement is available for figure 8:

**Figure supplement 1.** Gene ontology analysis of differentially expressed retinal genes induced by microglial TGFBR2 ablation in functional networks.

DOI: https://doi.org/10.7554/eLife.42049.017

specifically, TGFBR2 is expressed in retinal microglia and endothelial cells as shown here. As evidenced by these complex expression patterns, TGFβ-mediated interactions in the retina are diverse and context-dependent; for example, constitutive TGFβ signaling to endothelial cells maintains the structural stability of the choroidal and retinal vascular circulation (*Schlecht et al., 2017*; *Walshe et al., 2009*), while that to retinal ganglion neurons promotes their differentiation and survival (*Braunger et al., 2013*; *Walshe et al., 2011*). Specific control of TGFβ signaling across these disparate contexts may be enabled by the local nature of interactions, such as through direct cell-cell contact, as well as through the agency of 'milieu molecules', such as LRRC33, which can influence localized and selective activation of TGFβ in specific cell types (*Qin et al., 2018*).

In our study here, we investigated the specific regulatory influence of constitutive TGFβ signaling on microglia in the retina. In the healthy adult retina, microglia are spatially organized in regular, non-overlapping, horizontal arrays concentrated within the synaptic plexiform layers (*Santos et al., 2008*). This ordered organization of ramified cells, together with the dynamic motility of microglial processes, provides for comprehensive spatial coverage and microglia-synapse contact in the IPL and OPL (*Lee et al., 2008*), enabling the maintenance of retinal synapses; depletion of retinal microglia results in progressive synaptic degeneration which can be rescued by microglial repopulation (*Wang et al., 2016*; *Zhang et al., 2018*). Microglia distribution in the retina also shows laminar specificity, with the outer retina being uniformly devoid of microglia under healthy conditions. This exclusion is functionally significant as the infiltration of microglia into the outer retina, which occurs in aging (*Xu et al., 2008*) and photoreceptor injury (*Ng and Streilein, 2001*), is associated with deleterious changes to photoreceptors and RPE cells (*Combadière et al., 2007*; *Ma et al., 2009*). We found in the current study that constitutive TGFβ signaling to retinal microglia is necessary for the maintenance of this overall organization, with retinal microglia demonstrating progressive disorganization in number, morphology, and distribution with TGFBR2 ablation. It is likely that TGFβ signaling can serve as a direct signal for microglial homeostasis; also, the induction of microglial 'sensome' gene products can enable microglia to orient themselves to environmental guidance cues. With TGFBR2 ablation, the downregulation of key receptors such as *Cx3cr1* and *P2ry12* can render microglia less responsive to CX3CL1, which regulates microglial activation and distribution (*Carter and Dick, 2004*; *Combadière et al., 2007*), and to ATP which promotes microglial morphological ramification and dynamic behavior (*Fontainhas et al., 2011*; *Liang et al., 2009*).

We found that this loss of microglial organization in the absence of TGFβ signaling is highly consequential to the function of the retina, such as in its electric response to light stimuli which subserves vision. The induced loss in microglial ramification, which likely resulted in decreased microglia-synapse contact, was associated with significant decrements of synaptic function in the form of abnormal b-to-a wave ratios in ERG responses. The mechanisms underlying microglia-synapse maintenance, while incompletely understood, have been related to local microglia-mediated delivery of neurotrophic factors. For example, genetic ablation of microglia-specific BDNF expression in the brain inhibited learning-related synapse formation and decreased cognitive behavior (*Parkhurst et al., 2013*). In support of these mechanisms, we observed that TGFBR2-ablation in retinal microglia resulted in a downregulated *Bdnf* expression, while exogenous TGFβ ligand stimulation conversely increased it in cultured retinal microglia. TGFBR2 ablation in microglia also resulted in abnormal microglial displacement into the subretinal space, akin to that seen with aging, suggesting that aging-related decreases in microglial responses to TGFβ (*Rozovsky et al., 1998*) may be a mechanism contributing to the misdistribution of microglia in the senescent retina.

We observed that the loss of TGFβ signaling in retinal microglia was also accompanied by the induction of microglial proliferation, increased mRNA expression of microglial activation markers (*Cd68, Cd74*), antigen presentation molecules (*H2-Aa*), proinflammatory cytokines (*Ccl2, Ccl8*), and increased immunopositivity to microglial activation markers (CD68, MHCII, F4/80, CD74, CD45). As also noted in TGFBR2-deficient brain microglia (*Buttgereit et al., 2016*; *Lund et al., 2018a*), these

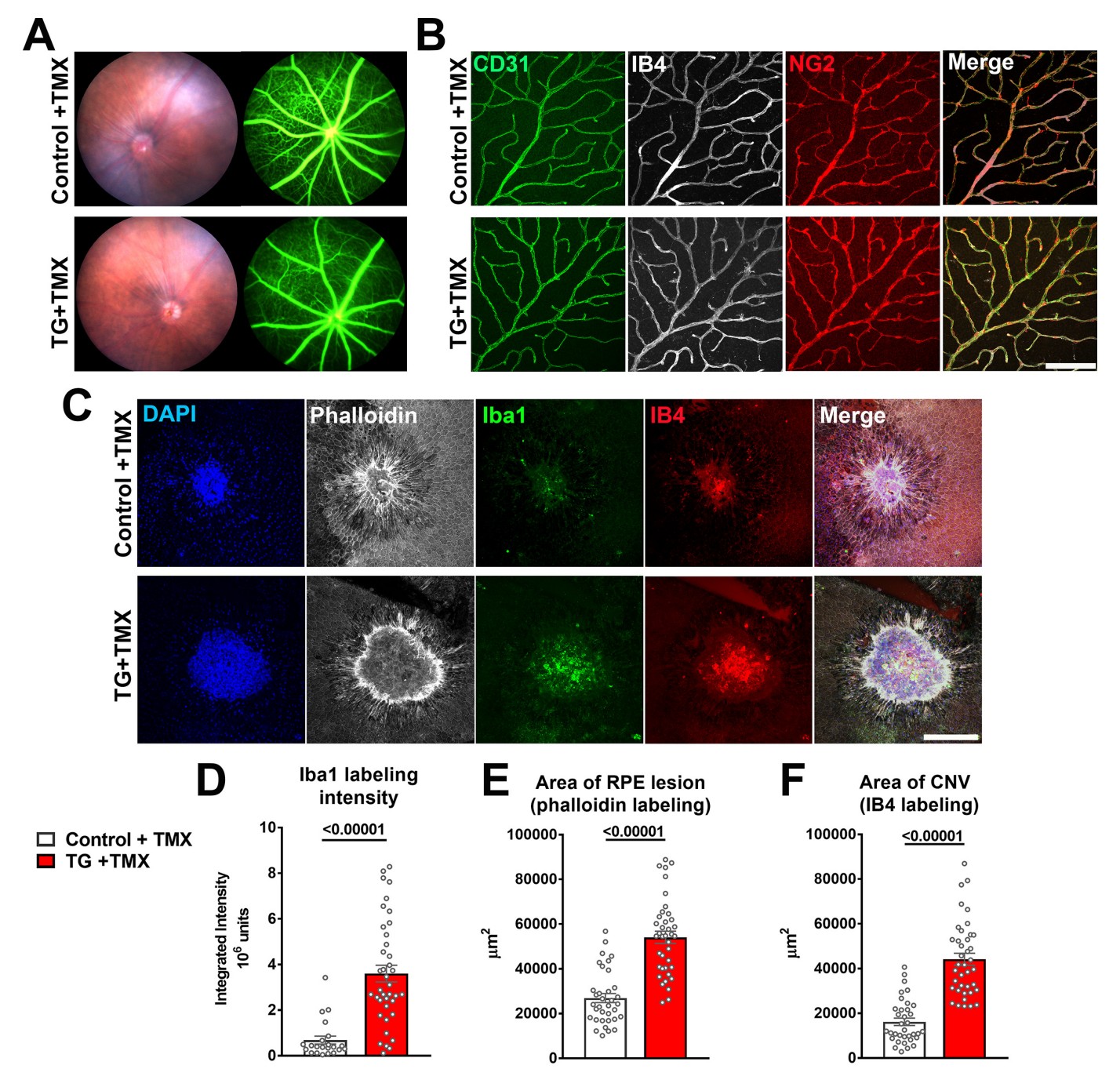

**Figure 9.** TGFBR2 ablation in retinal microglia increases pathological choroidal neovascularization (CNV) in an in vivo laser injury model. (**A**) *In vivo* evaluation for abnormalities in retinal vascular permeability using fluorescein angiography was performed in control and TG mice 12 weeks following tamoxifen (TMX) administration beginning at the age of 2 months. No abnormal leakage or vascular structure were detected. (**B**) Immunohistochemical analysis of endothelial cells (labeled with IB4 and an antibody to CD31) and retinal pericytes (labeled with an antibody to NG2) in retinal vasculature showed normal morphologies and distributions following TGFBR2 ablation in TG mice 12 weeks post-TMX. Scale bar = 100 µm. (**C**) Control and TG mice 3 weeks post-TMX were subjected to *in vivo* laser injury in a model of CNV formation. CNV complexes were analyzed in RPE flat-mounts using immunohistochemistry 7 days after laser injury and compared. Scale bar = 200 µm. TGFBR2-ablated TG animals demonstrated a higher recruitment of Iba1+ myeloid cells to the laser injury site (**D**), which was correlated with a larger laser lesion size (as labeled with phalloidin) (**E**) and a larger CNV area (**F**). (p values are from unpaired t-tests with Welch's correction, data points are from 40 lesions from six animals in each group).
DOI: https://doi.org/10.7554/eLife.42049.018

changes indicated a transition of resident microglia from a homeostatic to a more proinflammatory phenotype. Our experiments with cell-fate mapping and P2RY12-immunohistochemical analyses indicated that this alteration was predominantly constituted by a transition in resident microglia; we did not detect the infiltration of monocyte-derived macrophages, as was the case in comparable experiments examining the brain (*Lund et al., 2018a*). The reasons for why monocytic infiltration did not occur in this context are unclear and may be related to the absence of evident breakdown of the blood-retinal barrier, despite increased microglial activation, and also to the absence of an empty myeloid cell niche available to accommodate infiltrating monocytes (*Lund et al., 2018b*). It is possible that long-lived *Cx3cr1*-expressing perivascular macrophages resident within the retina may also contribute to the transformed population, but this is likely a smaller contribution, owing to their sparser numbers at baseline (*Goldmann et al., 2016*; *Mendes-Jorge et al., 2009*). We observed that retinal microglia following tamoxifen administration gradually acquired immunopositivity for CD206, a marker typically positive for perivascular macrophages. However, prominent proliferation and migration of CD206+ perivascular macrophages present at baseline prior to TGFBR2 ablation were not detected, indicating that true perivascular macrophages are unlikely to contribute substantially to the final population of transformed cells (*Figure 3—figure supplement 2*).

We found that this altered microglial phenotype induced by TGFBR2 ablation perturbed the regulation of neuroinflammation in the retina, which is mediated in part by microglia-Müller cell interactions (*Portillo et al., 2017*; *Wang and Wong, 2014*). We found evidence for widespread secondary gliotic changes in Müller cells that were spatiotemporally-coincident with microglial changes. Interestingly, these featured the selective upregulation of gliosis genes associated with neurotoxic reactive A1 astrocytes (*Liddelow et al., 2017*), indicating that a neurotoxic influence from Müller cells may be induced by TGFBR2-deficient microglia. These changes in Müller cells may additionally feedback onto nearby microglia via secreted signals to influence their activation (*Wang et al., 2011*; *Wang et al., 2014*). In addition, we found that TGFBR2 ablation resulted in a significant upregulation of *Apoe* in microglia, which has been linked with a reciprocal downregulation of TGFβ signaling in microglia, as well as an induction of a neurodegenerative microglial phenotype (*Krasemann et al., 2017*), that is also observed in profiling studies of brain microglia in models of aging and Alzheimer's disease (*Kang et al., 2018*). Correspondingly, we found that retinal microglia and Müller cells changes were accompanied by progressive retinal thinning and neuronal apoptosis, demonstrating that a deficiency in TGFβ-mediated microglial regulation results in dysregulated neuroinflammation that increases the vulnerability of the retina to neurodegenerative changes.

In a study published following the submission of our manuscript, Zöller et al., (*Zöller et al., 2018*) using a similar transgenic model, had induced the ablation of exon 2/3 of TGFBR2 in *Cx3cr1*-expressing cells (*Chytil et al., 2002*) and described in the brain an upregulation of microglial activation markers, but had failed to detect alterations in microglial density, microglia-specific gene expression or neuronal survival. This contrasts with our findings here and those in previous reports in which TGFBR2 ablation in adult microglia resulted in increased microglial proliferation and numbers, downregulation of microglia-specific genes such as *Siglech*, and the onset of behavioral phenotypes of degeneration (*Buttgereit et al., 2016*; *Lund et al., 2018a*). In Zöller et al., the stability in the expression of microglia-specific genes following TGFBR2 ablation may have arisen from experiments in which mRNA profiling was performed on microglia that had been sorted as CD45$^{low}$, which may have selected against TGFBR2-ablated microglia (which upregulate CD45 expression) and selected for the fraction of microglia that had not undergone gene recombination. The authors had assayed for neurodegenerative changes using only counts of NeuN+ cortical neurons, a method that is less sensitive than measures of neuronal apoptosis or assays of neuronal function for detecting neurodegenerative changes. Combined with findings *in vitro* demonstrating a requirement for TGF-β for the expression of a microglia-specific gene signature, and those *in vivo in* showing that decreased TGF-β signaling to microglia resulted in the downregulation of microglia-specific gene expression and neurodegenerative changes (*Butovsky et al., 2014*; *Qin et al., 2018*), it is likely that microglia in the retina require constitutive TGF-β signaling to maintain a microglia-specific gene signature and to prevent a transition to an activated phenotype that helps drive retinal neurodegeneration.

We found that in addition to regulating the homeostatic status of microglia, TGFβ signaling was also important in mediating injury-induced microglial responses in the retina. In a model of laser injury of choroidal neovascularization, we found that an increased number of TGFBR2-deficient microglia was recruited to the area of laser injury, which was associated with a greater area of RPE

disruption and an increased size of pathological CNV. Previous studies have found an association between CNV and recruited myeloid cells which can promote neovascularization by the expression of pro-angiogenic factors and inflammatory cytokines (*Crespo-Garcia et al., 2015*; *Li et al., 2017*); alterations in the polarization state of these myeloid cells were also influential in the extent of the CNV formed (*Kelly et al., 2007*; *Yang et al., 2016*). In this context, our observations posit a potential mechanism by which TGFβ signaling to microglia may regulate the level of chronic inflammation in the aged retina in pathologically significant ways, particularly with respect to AMD. The mechanism underlying the increased risk for AMD development in patients with polymorphisms associated with *Tgfbr1* (*Fritsche et al., 2013*) may relate to decreased levels of TGFβ signaling from the retinal environment to microglia in the AMD retinas, consequently shifting microglia from a homeostatic physiological state to a pathological state (*Figure 10*). This results in altered interactions between microglia and Müller cell and neurons, increasing the retina's vulnerability to synaptic and neuronal degeneration and exacerbating the severity of CNV development in response to pathogenic triggers, phenotypes that are both hallmarks of advanced AMD. Pathogenic mechanisms related to polymorphisms in *Htra1*, another gene associated with AMD risk, have also been related to alterations in the ability of HTRA1 to modulate TGFβ signaling in microglia (*Friedrich et al., 2015*). Also, in an amyloid-β-induced rodent model for AMD, intraocular delivery of exogenous TGFβ1 resulted in decreased markers of neuronal apoptosis (*Fisichella et al., 2016*), prompting proposals of modulation of TGFβ signaling as a potential AMD therapeutic strategy (*Platania et al., 2017*). Analogously, delivery of TGFβ ligands in rodent models of multiple sclerosis (*De Feo et al., 2017*) and hemorrhagic stroke (*Taylor et al., 2017*) has been found to facilitate immunomodulation of brain

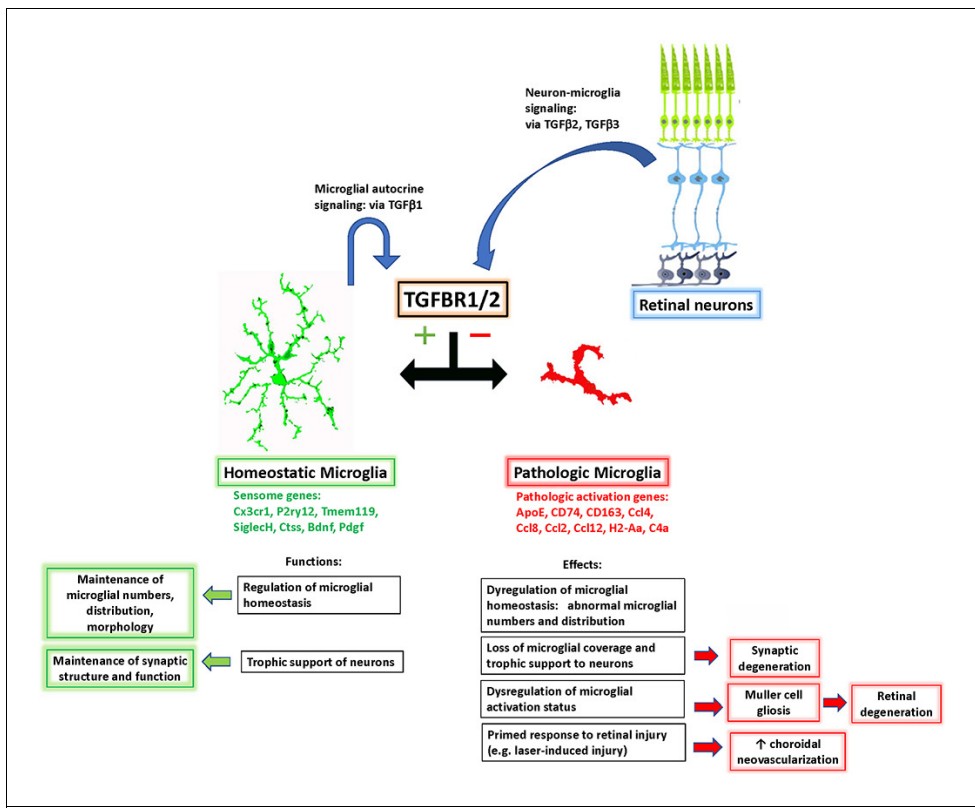

**Figure 10.** Schematic showing the role of TGFβ signaling in the regulation of retinal microglial physiology and the consequences of altered TGFβ signaling in the retina. TGFβ ligands, expressed constitutively by retinal neurons (TGFβ−2 and −3) and retinal microglia (TGFβ−1), signal to TGFBR2-expressing microglia to promote their homeostatic phenotype and to suppress a pathologic phenotype. Conversion between these phenotypes, which are associated with corresponding patterns of gene expression, results in a loss of microglial organization and microglial trophic functions and increased pathological neurodegeneration and neovascularization.
DOI: https://doi.org/10.7554/eLife.42049.019

microglia/macrophages to favor structural and functional neuronal recovery. As such, altered TGFβ signaling may constitute an important mechanism underlying the contribution of microglia to AMD pathobiology (*Guillonneau et al., 2017*).

Collectively, these results demonstrate that constitutive TGFβ signaling in retinal microglia is necessary in maintaining the organization of microglia in the healthy retina and is indispensable for their homeostatic function in synapse maintenance. This signaling is also necessary for the regulation of the neuroinflammatory status within the retina as insufficient levels results in aberrant microglial activation that drives chronic inflammatory changes, leading to a transformation of Muller cells to a maladaptive gliotic form, and an increased vulnerability of retinal neurons to neurodegeneration. TGFβ signaling also negatively regulates microglial responses to injury triggers without which exacerbated pathological choroidal neovascularization results. This combination of AMD-related phenotypes that involve chronic inflammation, neuronal degeneration, and pathological choroidal neovascularization, together with the genetic risk for AMD in TGFBR1 polymorphisms, implicate TGFβ regulation of retinal microglia as an influential contributing pathologic mechanism in AMD.

## Materials and methods

### Experimental animals

Experiments were conducted according to protocols approved by the National Eye Institute Animal Care and Use Committee and adhered to the Association for Research in Vision and Ophthalmology (ARVO) Statement for the use of animals in ophthalmic and vision research. Animals were housed in a National Institutes of Health animal facility under a 12 hr light/dark cycle with food *ad libitum*. Transgenic mice in which the *Cx3cr1* gene was replaced by a sequence encoding Cre recombinase with a tamoxifen-dependent estrogen ligand-binding domain and a downstream sequence for IRES-EYFP (*Cx3cr1 $^{CreER}$* mice, provided by Dr. Wen-Biao Gan, Skirball Institute) (*Parkhurst et al., 2013*) were crossed with mice possessing loxP sites that flank exon 4 of the transforming growth factor, beta receptor II (*Tgfbr2*) (*Tgfbr2$^{flox/flox}$*, The Jackson Laboratory, #012603) (*Levéen et al., 2002*) to generate *Cx3cr1$^{CreER/+}$,Tgfbr2$^{flox/flox}$* mice (termed TG mice) which enabled the inducible deletion of exon 4 of *Tgfbr2*. EYFP expression, as driven by the *Cx3cr1* promoter, was used as a marker for *Cx3cr1* expression. Transgenic mice in which one copy of the *Cx3cr1* gene was replaced by sequences coding for green fluorescent protein (GFP) (designated *Cx3cr1$^{+/GFP}$* mice, The Jackson Laboratory, #005582) (*Jung et al., 2000*) were used to label microglia. *Cx3cr1$^{CreER}$* mice were crossed with Ai14 mice harboring a loxP-flanked STOP cassette preventing the transcription of a CAG promoter-driven red fluorescent protein variant (tdTomato) inserted into the Gt(ROSA)26Sor locus (*Madisen et al., 2010*) (The Jackson Laboratory, #007914) constituted an alternative system to label microglia. These mice were also crossed with *Cx3cr1$^{CreER/+}$,Tgfbr2$^{flox/flox}$* mice to generate *Cx3cr1 $^{CreER/+}$,Tgfbr2$^{flox/flox}$*, Ai14/+ mice to enable cell-fate mapping of microglia vs. monocytes as previously performed (*Ma et al., 2017*). Cre recombinase activity was induced by tamoxifen administered by oral gavage (10 mg dose twice one day apart). *Tgfbr2$^{flox/flox}$* mice (termed 'Control' mice) which were administered tamoxifen on the same regimen served as controls to tamoxifen-administered TG mice. All experimental animals were genotyped by gene sequencing to confirm the absence of the rd8 mutation (*Mattapallil et al., 2012*).

### Immunohistochemical and TUNEL analysis of retinal tissue

Mice were euthanized by $CO_2$ inhalation and their eyes were removed. Enucleated eyes were dissected to form posterior segment eye-cups and fixed in 4% paraformaldehyde in phosphate buffer (PB) for 2–4 hr at 4°C. Eyecups were either cryosectioned (Leica CM3050S) or dissected to form retinal flat-mounts. Flat-mounted retinas were blocked for 1 hr in blocking buffer containing 10% normal donkey serum and 0.5% Triton X-100 in PBS at room temperature. Primary antibodies, which included IBA1 (1:500, Wako, #019–19741), TGFBR2 (1:100, R and D, #AF-241), CD31 (1:100, Bio-Rad, #MCA2388), GFAP (1:200, Invitrogen, #13–0300), TMEM119 (1:100, Abcam, #ab209064), CD68 (1:200, Biorad, #MCA1957), MHCII (I-A/I-E, 1:100, BD Bioscience, #556999), NG2 (1:200, Millipore, #05–710), CD74 (1:100, BD Biosciences, #555318), F4/80 (1:100, Bio-Rad, #MCA497), CD45 (1:100, Bio-Rad, #MCA1388), glutamine synthetase (1:200, Millipore, #MAB302), BRN3A (1:100, Santa Cruz, #SC31984), cone arrestin (1:200, Millipore, #AB15282), choline acetyltransferase (ChAT) (1:100,

Millipore, #AB144p), PKCα (1:200, Sigma-Aldrich, #p4334), CCL8 (1:100, Bio-Rad, #AAM62B), Ki67 (1:50, eBioscience, #50-5698-82), P2RY12 (1:100, ThermoFisher, #PA5-77671 and Sigma, #HPA014518), CD206 (1:100, BioRad, #MCA2235GA), were diluted in blocking buffer and applied overnight for sections and flat-mounts at room temperature on a shaker. Experiments in which primary antibodies were omitted served as negative controls. After washing in $1 \times$ PBST (0.2% Tween-20 in PBS), retinal samples were incubated for 2 hr at room temperature with secondary antibodies (AlexaFluor 488-, 568- or 633-conjugated anti-rabbit, mouse, rat or goat IgG) and DAPI (1:500; Sigma-Aldrich) to label cell nuclei. Isolectin B4 (IB4), conjugated to AlexaFluor 568/647 (1:100, Life Technologies), was used to label activated microglia and retinal vessels. Apoptosis of retinal cells was assayed using a terminal deoxynucleotidyl transferase dUTP Nick End Labeling (TUNEL) assay (in situ cell death detection kit, TMR red; Roche) according to the manufacturer's specifications. Stained retinal samples were imaged with confocal microscopy (Olympus FluoView 1000, or Zeiss LSM 880). For analysis at high magnification, multiplane z-series were collected using $20 \times$ or $40 \times$ objective; each z-series spanned from vitreal surface to the outer plexiform layer (OPL) for retinal flat-mounts, and over a depth of 20 μm for retinal sections, with each section spaced 1–2.5 μm apart. Confocal image stacks were viewed and analyzed with FV100 Viewer Software (Olympus), Zen software (Zeiss), and/or ImageJ (NIH).

## Isolation of retinal microglia by flow cytometry

Enucleated globes were immersed in ice-cold Hank's balanced salt solution (HBSS) and retinas were isolated by dissection before transfer into 0.2% papain solution including glucose (1 mg/mL), DNAse1 (100 U/mL; Worthington, Lakewood, NJ, USA), superoxide dismutase (SOD) (5 mg/mL; Worthington), gentamycin (1 μL/mL; Sigma), and catalase (5 mg/mL; Sigma, St Louis, MO, USA) in HBSS, and incubated at 8°C for 45 min and then at 28°C for 7 min. The digested tissue was dissociated by trituration and centrifuged at 150G for 5 min at 4°C. The resulting cell pellet was resuspended with neutralization buffer containing glucose (2 mg/mL), DNAse1 (100 U/mL), SOD (5 mg/mL), catalase (5 mg/mL), antipain (50 mg/mL Roche, Indianapolis, IN, USA), d-a-tocopheryl acetate (10 mg/mL; Sigma), albumin (40 mg/mL), and gentamycin (1 mL/mL, Sigma), and again centrifuged at 150 g for 5 min at 4°C. The cellular pellet was resuspended in 100 μL of staining buffer (catalog no. 554656, BD Pharmingen, San Diego, CA, USA) containing an Alexa Fluor 488-conjugated antibody to CD11b (1:50; catalog no. 53-0112-82, eBioscience, San Diego, CA, USA) and incubated for 20 min on ice. The cells were washed twice in 5 mL of staining buffer containing 2 mM ethylenediaminetetraacetic acid (EDTA) and suspended with 0.5 mL of staining buffer. Labeled retinal microglia were isolated by fluorescence-activated cell sorting (FACS)(BD FACSAria II Flow Cytometer; BD, Franklin Lakes, NJ, USA) at the NEI Flow Cytometry Core Facility. Sorted cells were collected into a 1.5 mL Eppendorf tube containing 200 μL of RNAlater solution (Ambion, AM7021) and stored at −80°C for subsequent RNA extraction.

## Quantitative PCR analysis

Quantitative PCR analysis of genomic DNA and mRNA from FACS-sorted cells was performed. Total RNA was extracted from sorted cells (RNeasy Mini kit, Qiagen, Valencia, CA, USA) and used to synthesize cDNA with MessageBooster cDNA synthesis kit (Epicentre) following the manufacturer's instructions and analyzed with qPCR in CFX96 real time PCR system (BioRad). Levels of mRNA expression were normalized to those in controls as determined using the comparative CT ($2\Delta\Delta CT$) method. Ribosomal protein S13 (RPS13) was used as an internal control. Oligonucleotide primer pairs used are listed in *Supplementary file 1*.

## mRNA profiling in retinal tissue using Nanostring

mRNA expression in retinal tissue was profiled and analyzed using the Nanostring platform nCounter Mouse Immunology panel containing 547 immunology-related mouse genes and 14 internal reference controls (Nanostring, Seattle, WA, #XT-CSO-MIM1-12). Briefly, the total RNA from a single retina was extracted using the RNeasy kit (Qiagen, Valencia, CA, USA). A total of 100 ng RNA in a volume of 5 μl was then hybridized to the capture and reporter probe sets at 65°C for 16 hr according to the manufacturer's instructions. The individual hybridization reactions were washed and eluted per protocol at a NIH Core Facility (CCR Genomics Core, NCI, Bethesda, MD) and the data collected

using the nCounter Digital Analyzer (Nanostring). Generated data was evaluated using internal QC process and the resulting data were normalized with the geometric mean of the housekeeping genes using the nSolver 4.0 and nCounter Advanced Analysis 2.0 software (Nanostring). Retina from 4 groups of animals, each comprising three biological repeats, were analyzed: control animals, not administered tamoxifen (Control), control animals, administered tamoxifen for 2 weeks (Control + TMX), TG animals, administered tamoxifen for 2 weeks (TG + TMX, 2 weeks), TG animals, administered tamoxifen for 8 weeks (TG + TMX, 8 weeks). Differentially expressed genes between two comparison groups were defined as those demonstrating a difference in expression level of fold change >2, with a p-value of <0.05 (adjusted p-value, t- test). The unsupervised hierarchical clustering analysis was performed using JMP statistical software (V13, SAS, Cary, NC). Canonical pathway analyses were performed using Ingenuity Pathway Analysis (IPA, Qiagen, Venlo, Netherlands).

## Retinal microglia cell culture

Retinal microglia were isolated from postnatal day (P)20 C57BL/6J wild type mice and heterozygous $Cx3cr1^{+/GFP}$ transgenic mice as previously described (*Ma et al., 2013*). Briefly, retinal cells were dissociated by digestion in 2% papain, followed by trituration and centrifugation. Resuspended cells were transferred into 75 cm$^2$ flasks containing Dulbecco's Modified Eagle Medium (DMEM): Nutrient Mixture F-12 media with 10% fetal bovine serum (FBS) (Gibco, Carlsbad, CA, USA) and nonessential amino acids solution (Sigma, St. Louis, MO, USA). Following overnight culture, the medium and any floating cells were discarded and replaced with fresh medium. When the cultures approached confluence and cells begin to show detachment, the culture flasks were shaken gently to detach microglial cells that were then subcultured in 6-well plates. When microglial cultures reached 60–70% confluence, they were exposed to TGFβ ligands for 6 or 24 hr and then harvested for mRNA analysis.

## In vivo optical coherence tomographic (OCT) imaging

Mice were anesthetized with intraperitoneal ketamine (90 mg/kg) and xylazine (8 mg/kg) and their pupils were dilated. Retinal structure was assessed using an OCT imaging system (Bioptigen; InVivo-Vue Software). Volume scans of 1.4 mm by 1.4 mm centered on the optic nerve (1000 A-scans/horizontal B-scan, 33 horizontal B-scans, average of three frames per B-scan, each spaced 0.0424 mm apart) were captured. Retinal thicknesses in each quadrant of a circular grid of diameter 1.2 mm were measured using the "measure" tool in the manufacturer's software. Total retinal thickness, measured from the nerve fiber layer to the retinal pigment epithelium (RPE) layer, and outer retinal thickness, measured from the outer plexiform layer to the inner surface of the RPE layer, were obtained. Inner retinal thickness was computed as the difference between total retinal thickness and outer retinal thickness.

## Electroretinographic (ERG) analysis

ERGs were recorded using an Espion E2 system (Diagnosys). Mice were anesthetized as described above after dark adaptation overnight. Pupils were dilated and a drop of proparacaine hydrochloride (0.5%; Alcon) was applied on cornea for topical anesthesia. Flash ERG recordings were obtained simultaneously from both eyes with gold wire loop electrodes, with the reference electrode placed in the mouth and the ground subdermal electrode at the tail. ERG responses were obtained at increasing light intensities over the ranges of $1 \times 10^{-4}$ to 10 cd/s/m$^2$ under dark-adapted conditions and 0.3 to 100 cd/s/m$^2$ under a rod-saturating background light. The stimulus interval between flashes varied from 5 s at the lowest stimulus strengths to 60 s at the highest ones. Two to 10 responses were averaged depending on flash intensity. ERG signals were recorded with 0.3 Hz low-frequency and 300 Hz high-frequency cutoffs sampled at 1 kHz. The a-wave amplitude was measured from the baseline to the negative peak and the b-wave was measured from the a-wave trough to the maximum positive peak. Statistical comparisons between tamoxifen-treated control and TG mice were analyzed using a two-way ANOVA.

## Image analysis

To analyze the total numbers of microglia in the retina, manual counts of Iba1+ cells were performed over the entire retina in flat-mounted specimens. Microglial counts, as well as the proportion of microglia that were Ki67+, were evaluated in the separate retinal lamina (IPL, OPL, and subretinal

space) using high-magnification image stacks captured at consistent retinal regions of interest (ROIs) positioned midway between the optic nerve and the retinal periphery. Morphological analysis of microglia on the parameters of number of branch points/cell and area of dendritic field were performed manually using Image J software (NIH). The intensity of immunohistochemical labeling was assessed by quantifying the mean fluorescent intensity within each labeled microglial cell using Image J software.

## Fluorescein angiography

Mice were injected intraperitoneally with Fluorescein AK-FLUOR (100 mg/mL; Akorn) at 100 µg/g (body weight). Fluorescein angiography (FA) of the retina was performed using a Phoenix Micron III retinal imaging system (Phoenix Research Labs) at various times following fluorescein injection. Bright-field and fluorescence images of the central fundus were captured during early and late transit phases.

## In vivo laser model for choroidal neovascularization

Choroidal neovascularization was induced in vivo using a laser injury model as previously described (*Campos et al., 2006*). Experimental animals were anesthetized with an intraperitoneal injection of ketamine (90 mg/kg) and xylazine (8 mg/kg) and their pupils were dilated with 1% tropicamide (Akorn Inc, Buffalo Grove, IL, USA) and 2.5% phenylephrine (Alcon Laboratories Inc, Fort Worth, TX, USA). Corneal anesthesia was provided using topical 0.5% proparacaine (Alcon Laboratories Inc). Laser injury was applied to the retina using a slit-lamp-mounted, 532 nm wavelength, photocoagulation laser (Iridex, Mountain View, CA, USA) and a handheld focusing lens. Using laser settings to create burns that ruptured Bruch's membrane (power, 50 mW; duration, 100 ms; spot size, 100 µm), four well-spaced laser burns were placed circumferentially approximately 375 µm from the optic nerve. Animals were sacrificed 7 days after laser injury and the size of choroidal neovascularization (CNV) complexes was evaluated in RPE-choroidal flat-mounts following immunohistochemical staining with DAPI, Iba1, Alex633-conjugated phalloidin (1:100), and Alex568-conjugated lectin IB4 (1:100). Microglial recruitment was evaluated by measuring the intensity of the Iba1+ signal, the area of RPE cell disruption was determined using phalloidin labeling, and the area of CNV determined by IB4 labeling and image analysis.

## Acknowledgements

This study is supported by funds from the National Eye Institute Intramural Research Program.

## Additional information

### Funding

| Funder | Grant reference number | Author |
|--------|------------------------|--------|
| National Eye Institute | Intramural Research Program | Wai T Wong |

The funders had no role in study design, data collection and interpretation, or the decision to submit the work for publication.

### Author contributions

Wenxin Ma, Conceptualization, Data curation, Formal analysis, Investigation, Methodology, Writing—original draft, Writing—review and editing; Sean M Silverman, Lian Zhao, Rafael Villasmil, Maria M Campos, Juan Amaral, Investigation, Writing—original draft, Writing—review and editing; Wai T Wong, Conceptualization, Resources, Data curation, Formal analysis, Supervision, Funding acquisition, Methodology, Writing—original draft, Project administration, Writing—review and editing

## Author ORCIDs

Wenxin Ma (iD) http://orcid.org/0000-0001-8396-6625
Lian Zhao (iD) http://orcid.org/0000-0002-0120-1969
Wai T Wong (iD) http://orcid.org/0000-0003-0681-4016

## Ethics

Animal experimentation: This study was performed in strict accordance with the recommendations in the Guide for the Care and Use of Laboratory Animals of the National Institutes of Health. All of the animals were handled according to approved institutional animal care and use committee (IACUC) protocols (NEI-602, NEI-665) of the National Eye Institute.

## Decision letter and Author response

Decision letter https://doi.org/10.7554/eLife.42049.026
Author response https://doi.org/10.7554/eLife.42049.027

## Additional files

### Supplementary files

• Supplementary file 1. Sequences of oligonucleotide primers used in polymerase chain reaction (PCR) assays
DOI: https://doi.org/10.7554/eLife.42049.020

• Supplementary file 2. Key resources table.
DOI: https://doi.org/10.7554/eLife.42049.021

• Transparent reporting form
DOI: https://doi.org/10.7554/eLife.42049.022

### Data availability

All data generated or analysed during this study are included in the manuscript and supporting files.

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
