## [Decision Letter]

Thank you for submitting your article "Absence of TGFβ signaling in retinal microglia induces retinal degeneration and promotes choroidal neovascularization" for consideration by *eLife*. Your article has been reviewed by three peer reviewers, and the evaluation has been overseen by a Reviewing Editor and Gary Westbrook as the Senior Editor. The following individuals involved in review of your submission have agreed to reveal their identity: Pat D'Amoore (Reviewer #2); Lois Smith (Reviewer #3). The reviewers have discussed the reviews with one another and the Reviewing Editor has drafted this decision to help you prepare a revised submission.

Summary:

The study, "Absence of TGFβ signaling in retinal microglia induces retinal degeneration and promotes choroidal neovascularization" addresses an interesting question in retinal degeneration, perhaps relevant to AMD and other degenerative diseases. Previous studies of the role of TGFb in retinal biology, and suggestions from human genetics, have led to the notion that microglia are signaled through this pathway. A role for microglia in retinal health has been addressed previously, including studies by this group, where they found that they play a role in rod degeneration in models of retinitis pigmentosa. Here, the authors deleted the TGFBR2 gene from mice using a conditional allele and a tamoxifen regulated Cre expressed from the regulatory region of Cx3CR1. They find that retinal neurons degenerate, there is a change in the response of microglia to laser injury, there is a reduction in the expression of microglial genes comprising their sensome, and overall a change in the microglial gene expression signature suggesting alterations to immune system modulation. The authors conclude that all of the observed phenotypes are due to effects within microglia following loss of the receptor.

Essential revisions:

As pointed out by the reviewers, the Cre allele that is used likely can lead to recombination in two other cell types, macrophages and monocytes. The authors should attempt to determine if recombination is solely within microglia by using antibodies to a microglial-specific antibody to Tmem119, and one to P2RY12. If they are unable to prove their statements regarding the sole recombination within microglia, they should modify their title and Discussion to reflect these ambiguities. However, the study is well done and will stand without this proof, but modifications of their claims will need to be made accordingly. Several other suggestions are made by the reviewers that should be addressed to improve the manuscript, e.g. a discussion of the source of the ligand and reference to the new work from Zoller et al. using a similar model to examine effects in the brain.

Original reviews:

The original reviews from the reviewers are below. We think addressing other issues that are not explicit in the Essential revisions above would also improve the manuscript so we hope you will consider them as you prepare your revisions.

Reviewer #1:

In the manuscript entitled "Absence of TGFβ signaling the retinal microglia induces retinal degeneration and promotes choroidal neovascularization", the authors identify that loss of TGFβR2 in retinal microglia influences microglial activation and neurodegeneration. Previous work has shown that CNS deletion of TGFβ1 results in loss of a homeostatic gene signature in microglia and a shift to a heightened inflammatory state (Butovksy et al., 2014). The authors take this a significant step further using conditional genetic deletion of TGFβR2 specifically in microglia (Cx3cr1^CreERT2^;Tgfbr2^fl/fl^ mice) and show that this results in reduced expression of homeostatic genes and a heightened inflammatory state in the retina. Importantly, the authors further show that this ultimately results in structural and functional degeneration of neurons in vivo in the retina. Overall, the authors address an important question with rigor. The data are also high quality and largely support the conclusions. The following are points that require further attention:

1) The authors do not address the possibility that the effects are due to changes in perivascular macrophages vs. retinal microglia. It is highly likely that PVMs also recombine using the Cx3cr1^CreERT2^ mouse. It would be important to address the potential contribution of these cells given their high association with blood vessels.

2) The contribution of infiltrating monocytes to increased numbers of Iba-1/CD11b-positive cells in Cx3cr1^CreERT2^;Tgfbr2^fl/fl^ mice has not been ruled out. Using tdTomato as a reporter is not sufficient (Figure 3—figure supplement 1C). While monocytes should turn over following tamoxifen treatment to eventually become tdTomato-negative (within ~7-14 days), tdTomato-positive monocytes could have infiltrated and taken up residence in the retina prior to turning over. One would not be able to distinguish these early infiltrates in the current experiments.

3) The authors likely submitted the current manuscript prior to the publication of the recent paper (October 1, 2018) by Zöller et al. in Nature Communications (Zöller et al., 2018). This group uses the same genetic strategy to assess TGFβ signaling in microglia (Cx3cr1^CreERT2^;Tgfbr2^fl/fl^ mice), but assessments were perform in the brain or in vitro. Interestingly, while the two studies observe similar results in heightened microglial inflammatory state following ablation of TGFβR2, the study by Zöller et al. does not observe decreases in microglial homeostatic gene expression or neurodegeneration. These differences in the two studies should be addressed in the revised manuscript.

4) It is concluded that effects on microglial morphology are cell autonomous (subsection “Specific in vivo ablation of TGFBR2 in retinal microglia induces rapid morphological transformation and proliferation”, first paragraph). While certainly a possibility, it is also plausible that these effects are secondary. For example, loss of TGFβ signaling in microglia could result in changes in signaling to Müller glia. The Müller glia then respond by releasing factors that alter the microglial morphology. Such alternative possibilities should be addressed.

5) One of the major conclusions is that microglia downregulate their 'sensome' related genes following ablation of TGFβR2. However, in Figure 7, the mutant microglia appear to react more robustly to a laser ablation? Purinergic signaling, a key sensome-related pathway, is known to regulate this injury response in microglia and many of these genes appear to be downregulated in the gene expression data set presented in this paper. Therefore, it is curious that such a robust response is observed in Figure 7. This should be discussed.

6) It seems inappropriate to include graphs of published data from other groups (Figure 1B and 1C). Instead, the authors should simply refer to these data in the text. Also, note that the Barres lab RNAseq database is largely from neonate animals, not adult.

7) The same image is used for 2 figures (3 week timepoint in Figure 2 and Figure Supplementary Figure 1). This should be corrected.

8) The image of Iba1 staining in Figure 3—figure supplement 1A at 3 weeks post-tamoxifen treatment is inconsistent with the increased Iba1 and CD11b-positve cells shown in Figure 2 and Supplementary figure 1. This should be addressed.

9) In Figure 2E and G, the authors would be better served to quantify the total area of microglia vs. the area 'subtended' by microglia. It is difficult to make such a measurement in static images and is better suited for live imaging. For example, perhaps mutant microglia are more motile and still 'subtend' the same amount of retina, but this can't be captured by static imaging.

10) It appears that microglia no longer express Cx3cr1 and Tmem119 in Figure 3A? It would be helpful to show these data on a graph where the y axis is a log2 scale.

11) CD68-positive lysosomes should be observed in control conditions, albeit at lower amounts compared to more activated cells. It is unclear why CD68 is virtually absent in controls (e.g. Figure 4C and D).

Reviewer #2:

While the work is novel for the retina, it is very reminiscent of a recent study from Qin et al., 2018, which shows that lack of activated TGFβ leads to microglial activation and progressive loss of myelinated axons and death. In addition, a paper was just published (Zoller et al., 2018) showing a similar role for TGFβ in brain microglia as in retina, which the authors may want to cite (unlikely they were aware of it prior to submission of this paper).

CX3CR1 is not specific for microglia and is also expressed by circulating monocytes. There is, however, one marker Tmem119 that has shown to be specific, which could provide "cleaner" results. A discussion of this fact would be useful. It is surprising that in spite of the demonstrated upregulation of proinflammatory cytokines, such as interferons and IL1β, by the activated microglial, infiltrating macrophages were not detected. The authors might speculate on why this is.

The authors conclude that the "constitutive neuron-microglia interactions in the form of TGFβ are necessary in the maintenance of the orderly organization and trophic function of the microglial in the retina". What is the evidence that neurons are providing the TGFβ, how is it activated and which isoform of TGFβ is mediating the homeostatic suppression of microglial activation?

The title is misleading. The data show that the absence of TGFβ signaling in microglial leads to exacerbated choroidal neovascularization, but the title suggests that this initiates the vessel growth.

Reviewer #3:

Inflammation/immune changes are associated with age-related macular degeneration (AMD). The authors examine how TGFβ signaling in microglia influences retinal neuroinflammation. They found that ablation of the TGFβ receptor, TGFBR2, in retinal microglia of adult mice induced abnormal microglial numbers, distribution, morphology, and activation status, and promoted a change in microglial gene expression profile. TGFBR2-deficient retinal microglia induced secondary gliotic changes in Müller cells, neuronal apoptosis, and decreased light-evoked retinal function (ERG) reflecting abnormal synaptic transmission and increased laser-induced choroidal neovascularization. These results suggest that TGFβ-mediated microglial regulation can drive neuroinflammatory contributions to AMD-related neurodegeneration and neovascularization.

General comments:

There is a dearth of cell specific information about "inflammation" and "immune" contributions to AMD. Although there are very clear indications of the importance of inflammation and immunity from human genetic studies, the cellular and biochemical details need to be worked out experimentally. There are many intrinsic problems with extrapolation of mouse data to human as the immune systems in each are very different and mouse models of AMD are approximate as mice have no macula, and aging in mice and humans is very different. Nonetheless, pathway analysis can be helpful. It is likely that these mouse analyses of TGFβ in macrophages/microglia will overlap with human AMD.

In many ways this paper is quite descriptive as there is no explanation for the neuronal loss following macrophage loss of TGFbR2. Nonetheless, the paper adds important new general information about macrophage neuronal interactions and TGFb control of macrophage function and related neuronal survival. I expect that with the techniques now available to look at multiple pathways the need to prove in detail that one pathway is involved after a genetic change in a cell will become less important.

Specific comments:

Figure 1B: need to label y axis as Tgfbr2 and reference the control. Check all figures for more complete labeling with units and specific mRNAs etc.

Figure 1 convincingly shows that TGFBR2 is expressed preferentially in microglia and is ablated in Cx3Cr1^CreER/+^,Tgfbr2^flox/flox^ mice with Tamoxifen.

Figure 2: TGFBR2 ablation in retinal microglia clearly induces abnormalities in microglial density, distribution, and morphology. Interestingly, despite increased density, TMX-treated TG retinas have a greater area without microglial processes.

Figure 3: Constitutive expression of microglial "sensome" genes is decreased with TGFBR2 ablation in retinal microglia. mRNA levels of Cx3cr1, P2yr12, Tmem119, and Siglech are all decreased in microglia in TG vs. control mice.

Figure 4: Expression of genes associated with microglial activation is increased with TGFBR2 ablation in retinal microglia. mRNA levels for H2-Aa (MHCII), Cd68, Cd74, Apoe, Ccl2, and CCl8 are all increased in microglia in TG vs. control mice.

Figure 5: TGFBR2 ablation in retinal microglia preserved lamination but progressively decreased both the inner and the outer retinal layer thickness (the inner plexiform layer (IPL), inner nuclear layer (ONL), outer plexiform layer (OPL), and outer nuclear layer (ONL)) 3 weeks post-TMX.

In 10 weeks post-TMX, TG animals had altered dark-adapted responses with a small decrease in a-wave amplitude and a marked decrease in b-wave amplitude. Light-adapted responses were similar for a-wave amplitude but significantly decreased in b-wave amplitude.

Figure 6: Changes in the mRNA expression of immune regulated genes in the retina following microglial TGFBR2 ablation using Nanostring-based profiling. Volcano plots at 2 and 8 weeks reflect the activation of neuroinflammatory pathways and immune cell activation and maturation pathways.

Figure 7: TGFBR2 ablation in retinal microglia increases laser-induced choroidal neovascularization. No abnormal leakage or vascular structure were detected. Normal morphologies and distributions following TGFBR2 ablation in TG mice 12 weeks TGFBR2-ablated TG animals demonstrated a higher recruitment of Iba1+ myeloid cells to the laser injury site, which was correlated with a larger laser lesion size and a larger CNV area.

---

## [Author Response]

Essential revisions:As pointed out by the reviewers, the Cre allele that is used likely can lead to recombination in two other cell types, macrophages and monocytes. The authors should attempt to determine if recombination is solely within microglia by using antibodies to a microglial-specific antibody to Tmem119, and one to P2RY12. If they are unable to prove their statements regarding the sole recombination within microglia, they should modify their title and Discussion to reflect these ambiguities. However, the study is well done and will stand without this proof, but modifications of their claims will need to be made accordingly.

In this revision, we have performed additional experiments to further support the notion that recombination (that enables TGFBR2 genetic ablation) occurs within retinal microglia by using microglia-specific antibodies as the reviewers had indicated. In our original manuscript, we had found that retinal transcripts for *Tmem119* and *P2ry12* declined to low levels 2 weeks following tamoxifen induction in transgenic animals. We had also found that TMEM119 immunopositivity in microglia is rapidly lost upon induction (as previously shown in Figure 3E, now Figure 4E). In this revision, we performed experiments with P2RY12 immunohistochemistry that demonstrated that during the first week following induction, all endogenous CD11b+, P2RY12+ retinal microglia demonstrated progressive deramification and morphological changes; many of these deramifying CD11b+, P2RY12+ cells also acquired Ki67+ immunopositivity. We did not observe any CD11b+, P2RY12-negative infiltrating macrophages entering the retina during this early phase. Although P2RY12 immunopositivity, like that for TMEM119, gradually declined and became undetectable after >7 days following induction, there is no evidence for a contribution of infiltrating monocytes during this early phase to the final population of morphologically transformed cells. Our cell-fate mapping experiments also ruled out the possibility for a later phase of monocyte infiltration as a contributing population. Because Ki67+ immunopositivity can also be detected in P2RY12+ microglia following induction, proliferation of endogenous microglia is a mechanism that is present and capable of contributing to the increased numbers of myeloid cells detected without the need to invoke a contribution from infiltrating macrophages.

These results, together with supporting data from Lund et al., 2018 (PMID: 29662171) and Buttgereit et al., 2016 (PMID: 27776109) (elaborated below in responses to reviewer #1), provide further justification for a microglial, rather than a monocytic, source for the morphologically transformed cells following TGFBR2 ablation.

This new data is included in Figure 3—figure supplement 1. We have added to the Results:

“In addition, we observed that the myeloid cells in the retina demonstrating progressive morphological change in the first week following tamoxifen administration were immunopositive for P2RY12, a marker for endogenous microglia, as well as for Ki67, a marker of proliferating cells (Figure 3—figure supplement 1D). Although P2RY12 immunpositivity was gradually lost after one week following TGFBR2 ablation, these findings indicated that the population of morphologically-transformed myeloid cells in the retina arose from the proliferation and modification of pre-existing endogenous retinal microglia.”

Several other suggestions are made by the reviewers that should be addressed to improve the manuscript, e.g. a discussion of the source of the ligand and reference to the new work from Zoller et al. using a similar model to examine effects in the brain.

We have in this revision: (1) added a new paragraph in the Discussion discussing the work of Zoller et al., (2) added further clarification regarding potential cellular sources of TGFB ligands in the retina.

Original reviews:The original reviews from the reviewers are below. We think addressing other issues that are not explicit in the Essential revisions above would also improve the manuscript so we hope you will consider them as you prepare your revisions.Reviewer #1:[…] The following are points that require further attention:1) The authors do not address the possibility that the effects are due to changes in perivascular macrophages vs. retinal microglia. It is highly likely that PVMs also recombine using the Cx3cr1^CreERT2^ mouse. It would be important to address the potential contribution of these cells given their high association with blood vessels.

The reviewer raises a valid point regarding the potential contribution of perivascular macrophages to the transformed population of myeloid cells within the retina following TGFBR2 ablation. While much remains unknown about perivascular macrophages, it is thought that they are a sparse population of perivascular cells located outside the retinal neural parenchyma (PMID: 19608545), that are long-lived and CX3CR1-expressing (PMID: 27135602). While we cannot rule out that there may be some transformed perivascular macrophages in the retina following TGFBR2 ablation, we consider it more likely that the majority of the effects described here are attributable to TGFBR2-ablated retinal microglia for the following reasons: (1) retinal microglia are much more numerous in the retina, (2) we observed progressive, time-dependent changes following tamoxifen administration in IBA1+ microglia located in the retinal neural parenchyma, which transitioned progressively from a ramified to an elongated morphology following TGFBR2 ablation. It is likely that CX3CR1-expressing perivascular macrophages undergo TGFBR2 ablation also, but we believe that the contribution to overall retinal pathology observed is likely small, given their sparse numbers at baseline. The oft-used marker for typical perivascular macrophages, CD206, is not helpful in making this distinction as retinal microglia, following TGFBR2 ablation, also become CD206-immunpositive. We performed additional experiments in which we conducted close follow-up of retinal microglia during the first week following tamoxifen ablation; we found that microglia demonstrated progressive morphological changes that were coincident with the gradual increase of CD206 immunopositivity in these cells. In order for true perivascular macrophages (which are CD206+ at baseline) to contribute substantially to the final population of myeloid cells in the TGFBR2-ablated retina, these cells will have to proliferate rapidly and migrate to distribute themselves across the retina – these features were not observed on close scrutiny of early events following TGFBR2 ablation.

We have added the following statement to the Discussion to include mention of perivascular macrophages:

“It is possible that long-lived CX3CR1-expressing perivascular macrophages resident within the retina may also contribute to the transformed population, but this is likely a smaller contribution, owing to their sparser numbers at baseline (Goldmann et al., 2016; Mendes-Jorge et al., 2009). […] However, prominent proliferation and migration of CD206+ perivascular macrophages present at baseline prior to TGFBR2 ablation were not detected, indicating that true perivascular macrophages are unlikely to contribute substantially to the final population of transformed cells (Figure 3—figure supplement 2).”

2) The contribution of infiltrating monocytes to increased numbers of Iba-1/CD11b-positive cells in Cx3cr1^CreERT2^;Tgfbr2^fl/fl^ mice has not been ruled out. Using tdTomato as a reporter is not sufficient (Figure 3—figure supplement 1C). While monocytes should turn over following tamoxifen treatment to eventually become tdTomato-negative (within ~7-14 days), tdTomato-positive monocytes could have infiltrated and taken up residence in the retina prior to turning over. One would not be able to distinguish these early infiltrates in the current experiments.

We agree with the reviewer that if monocytes did indeed infiltrate the retina soon after tamoxifen administration, subsequently becoming long-lived cells, they may, like endogenous microglia, demonstrate tdTomato labeling in this experiment. Our experimental design would rule out ongoing monocytic infiltration occurring after 7-14 days but may miss early infiltration. In this revision, we have provided additional data that show that all retinal microglia demonstrating morphological changes during this early period were initially immunopositive for P2RY12, a marker for endogenous microglia, supporting the notion that resident retinal microglia undergo morphological transformation to form the population of myeloid cells characterized in the manuscript. We did not observe any early infiltration of P2RY12-negative monocytes into the retina during this period.

The absence of a contribution by infiltrating monocytes in this context is additionally supported by previously published work: (1) Lund et al., 2018 (PMID: 29662171) in which bone-marrow transplantation from a CD45.1 WT donor mouse into a CD45.2, Cx3cr1^CreER/+^, Tgbr2^fl/fl^ recipient mouse demonstrated that all the CD11b+, F4/80+ myeloid cells located in the brain following TGFBR2 ablation were all CD45.2+, indicating their status as resident microglia (with little or no contribution from CD45.1+ circulating monocytes), (2) Buttgereit et al., 2016 (PMID: 27776109), in which tamoxifen administration in Sall1^CreER/+^, Tgbr2^fl/fl^, R26-YFP mice in resulted in TGFBR2 ablation and YFP expression specifically in only microglia and not circulating monocytes (Sall1 being specifically expressed in microglia but not other members of the mononuclear phagocyte family, such as monocytes); all the CD45+,F4/80+ cells in the brain following TGFBR2 were YFP+, indicating minimal contribution by systemic monocytes. We have in this revision also added new data showing that morphologically transformation myeloid cells in the retina following TGFBR2 ablation are immunopositive for P2RY12, a marker of endogenous microglia (see response to Essential revisions above).

The evidence that we have demonstrated in the retina, combined with other supporting information from previous studies in the brain, together indicate that monocytic contribution to the population of TGFBR2-ablated myeloid cells in the CNS, is unlikely to be significant.

3) The authors likely submitted the current manuscript prior to the publication of the recent paper (October 1, 2018) by Zöller et al. in Nature Communications (Zöller et al., 2018). This group uses the same genetic strategy to assess TGFβ signaling in microglia (Cx3cr1^CreERT2^;Tgfbr2^fl/fl^ mice), but assessments were perform in the brain or in vitro. Interestingly, while the two studies observe similar results in heightened microglial inflammatory state following ablation of TGFβR2, the study by Zöller et al. does not observe decreases in microglial homeostatic gene expression or neurodegeneration. These differences in the two studies should be addressed in the revised manuscript.

We have now included mention of this recent paper which had been published after our manuscript’s submission in the Discussion. While there are similarities between the findings in Zöller et al., and those in our manuscript and the work of others, there were also observations in their report that were not found elsewhere. They did not find in their observations evidence of decreased microglial homeostatic gene expression or the onset of neurodegenerative changes; this contrasted with not only our findings, but also those reported previously. We feel that these differences may have arisen in differences in methodology, and we remain in the opinion that microglia in the retina require constitutive TGF-β signaling to maintain a microglia-specific gene signature and to prevent a transition to an activated phenotype that help drive retinal neurodegeneration. We have added the following text in the Discussion to address these differences:

“In a study published following the submission of our manuscript, Zöller et al., (Zoller et al., 2018) using a similar transgenic model, had induced the ablation of exon 2/3 of TGFBR2 in CX3CR1-expressing cells (Chytil et al., 2002) and described in the brain an upregulation of microglial activation markers, but had failed to detect alterations in microglial density, microglia-specific gene expression or neuronal survival. […] Combined with findings in vitro demonstrating a requirement for TGF-β for the expression of a microglia-specific gene signature, and those in vivo in showing that decreased TGF-β signaling to microglia resulted in the downregulation of microglia-specific gene expression and neurodegenerative changes (Butovsky et al., 2014; Qin et al., 2018), it is likely that microglia in the retina require constitutive TGF-β signaling to maintain a microglia-specific gene signature and to prevent a transition to an activated phenotype that helps drive retinal neurodegeneration.

4) It is concluded that effects on microglial morphology are cell autonomous (subsection “Specific in vivo ablation of TGFBR2 in retinal microglia induces rapid morphological transformation and proliferation”, first paragraph). While certainly a possibility, it is also plausible that these effects are secondary. For example, loss of TGFβ signaling in microglia could result in changes in signaling to Müller glia. The Müller glia then respond by releasing factors that alter the microglial morphology. Such alternative possibilities should be addressed.

We had suggested that neuronal degeneration observed in our model may have been contributed by changes in secreted signals from Müller glia. We add to this point now by mentioning how these signals can also feedback onto microglia to influence them. The section in the Discussion now reads:

“We found evidence for widespread secondary gliotic changes in Müller cells that were spatiotemporally-coincident with microglial changes. […] These changes in Müller cells may additionally feedback onto nearby microglia via secreted signals to influence their activation (Wang et al., 2011; Wang et al., 2014).”

5) One of the major conclusions is that microglia downregulate their 'sensome' related genes following ablation of TGFβR2. However, in Figure 7, the mutant microglia appear to react more robustly to a laser ablation? Purinergic signaling, a key sensome-related pathway, is known to regulate this injury response in microglia and many of these genes appear to be downregulated in the gene expression data set presented in this paper. Therefore, it is curious that such a robust response is observed in Figure 7. This should be discussed.

The functional concept of expression of “sensome” genes by microglia in the healthy CNS has been related to the ability of microglia to recognize endogenous ligands in order to carry out homeostatic functions under normal conditions. In the absence of TGF-β signaling, the decrease in microglia-specific gene expression, and the increase in microglial activation markers and microglial proliferation, have been correlated to increased numbers of microglia demonstrating abnormal physiologies and aberrant responses that fail to successfully re-establish homeostasis in the face of perturbations, driving increased pathological change in the brain and spinal cord (Lund et al., 2018a, Qin et al., 2018, and Taylor et al., 2017).

In the retina, we also find that constitutive TGFβ signaling negatively regulates microglial responses to injury triggers, as TGFBR2 ablation results in more microglial recruitment and increased neovascular changes. While the decreased expression of P2RY12, a microglial receptor and sensome gene, may negatively influence microglial recruitment in response to ATP release induced by tissue injury, this single factor, in and of itself, does not necessarily predict a less robust microglial response. On the contrary, in the absence of TGF-β signaling, the increased numbers of microglia expressing higher levels of multiple priming and activation markers, culminate in increased inflammation and greater pathological change. Consistent with this, increased numbers of activated, proinflammatory microglia have been associated with increased pathological neovascularization in the retina (PMID: 29654250, 29767277, 23977372) – in the current context, proinflammatory changes in microglia induced by TGFBR2 ablation appear more influential to exacerbation of laser-induced neovascularization, than the potential loss of P2RY12 function. Our Nanostring mRNA profiling experiments detected in TGFBR2-ablated microglia increased expression of multiple chemokine receptors (CCR3, CCR5, CCR6) and Toll-like receptors (TLR1, TLR4, TLR9) that suggest an enhanced ability to respond to other inflammatory cues.

6) It seems inappropriate to include graphs of published data from other groups (Figure 1B and 1C). Instead, the authors should simply refer to these data in the text. Also, note that the Barres lab RNAseq database is largely from neonate animals, not adult.

We appreciate the reviewer’s point of view but we feel that the graphical representations of this data (which is other located within the published database and needs to be manually retrieved) help to communicate our point to the reader better than simply referring to it in the text. We have made the sources of the information very clear in the manuscript (there is no doubt as to its attribution). We have also changed the heading in Figure 1C (data from the Barres lab) from “adult” to “P7-17”.

7) The same image is used for 2 figures (3 week timepoint in Figure 2 and Supplementary Figure 1). This should be corrected.8) The image of Iba1 staining in Figure 3—figure supplement 1A at 3 weeks post-tamoxifen treatment is inconsistent with the increased Iba1 and CD11b-positve cells shown in Figure 2 and Supplementary figure 1. This should be addressed.

We apologize for inadvertently using the same image in two places in the manuscript. While these images are illustrative of the data and the point made in the manuscript, we have now provided alternative images in Supplementary Figure 1 (now Figure 2).

9) In Figure 2E and G, the authors would be better served to quantify the total area of microglia vs. the area 'subtended' by microglia. It is difficult to make such a measurement in static images and is better suited for live imaging. For example, perhaps mutant microglia are more motile and still 'subtend' the same amount of retina, but this can't be captured by static imaging.

In our analyses, we had aimed to quantitate morphological measures that provide a comprehensive and insightful description of morphological changes that microglia undergo on TGFBR2 ablation. One of the features that microglia have in the flat, laminated structure of the healthy adult retina, is in their two-dimensionally oriented, ramified morphologies that provide spatial coverage in the OPL and IPL. With TGFBR2 ablation, the ramified processes are shortened, the geometries of the cells altered, and the spaces between neighboring cells not covered by the processes enlarge. We had attempted to illustrate this feature by quantifying area subtended by the territory of each microglia’s dendritic tree. This difference in space occupation between cell morphologies with and without TGFBR2 ablation will not be highlighted by simply quantifying the total area of the microglial processes. This morphological analysis had been used in previous publications by us (Ma et al., 2009 and PMID: 21108733) and others (PMID: 25064005).

10) It appears that microglia no longer express Cx3cr1 and Tmem119 in Figure 3A? It would be helpful to show these data on a graph where the y axis is a log2 scale.

The main point of this figure is to illustrate that mRNA levels for microglial sensome genes are markedly downregulated following TGFBR2 ablation. We feel that the representation of the same data on a log2 scale may not reflect this main point as intuitively (see Author response image 1). In Figure 4A, we use the same graphical format to show the upregulation of microglial activation genes – in this situation, a matching log2 scale will be even less intuitive and evocative of the main point. This is our point of view concerning the effective exposition of our data and we are amenable to changing the format of the graph if there is consensus opinion among the editors and reviewers that a change is preferable.

11) CD68-positive lysosomes should be observed in control conditions, albeit at lower amounts compared to more activated cells. It is unclear why CD68 is virtually absent in controls (e.g. Figure 4C and D).

CD68-positive lysosomes are indeed visible in control microglia in the figure (now Figure 5C). At higher magnification, it can be observed that this signal is present in microglia in the expected locations (i.e. within the soma in a perinuclear location, close to the base of a primary process).

Reviewer #2:While the work is novel for the retina, it is very reminiscent of a recent study from Qin et al., 2018, which shows that lack of activated TGFβ leads to microglial activation and progressive loss of myelinated axons and death. In addition, a paper was just published (Zoller et al., 2018) showing a similar role for TGFβ in brain microglia as in retina, which the authors may want to cite (unlikely they were aware of it prior to submission of this paper).

We thank the reviewer for pointing out the paper Zoller et al., – it had not been published upon our submission. In this revision we had included mention of the data in these two very recent publications, including a detailed description of the findings in Zoller et al., (please see Discussion and comments above).

CX3CR1 is not specific for microglia and is also expressed by circulating monocytes. There is, however, one marker Tmem119 that has shown to be specific, which could provide "cleaner" results. A discussion of this fact would be useful.

Tmem119 has been described as a marker that is expressed in parenchymal microglia in the steady state, and not by non-CNS macrophages or by circulating monocytes. However, with the ablation of microglial TGFBR2, there is a downregulation of Tmem119 expression in microglia on a mRNA and protein level that makes long-term tracking of these cells difficult in this context (now Figure 4E). To address the reviewer’s suggestion however, we have performed additional experiments with another microglia-specific marker, P2RY12 (experiments described under “Essential revisions”) as part of this revision that provide further support to the notion that altered IBA1+ cells in the retina following TGFBR2 ablation originate from resident retinal microglia (that express the specific microglial marker, P2RY12), rather than from circulating monocytes.

It is surprising that in spite of the demonstrated upregulation of proinflammatory cytokines, such as interferons and IL1β, by the activated microglial, infiltrating macrophages were not detected. The authors might speculate on why this is.

It is not completely clear why circulating monocytes following TGFBR2 ablation do not infiltrate into the retina despite the upregulation of microglial activation factors. This however appears to be a finding that is also observed in the brain and spinal cord. We do not know with certainty the reasons why this is the case but we are happy to speculate. One possibility that is that infiltration of monocytes in various contexts are accompanied by alterations/breakdown in the blood-retina (or blood-brain) barrier, a feature that not evident the situation here. Another factor is the availability of an empty myeloid cell niche within the retina. In our recently published data (Zhang et al., 2018), we find that microglial repopulation in the retina following depletion also involve little monocytic infiltration, as endogenous microglia appear to proliferate sufficiently and rapidly enough to fill up the vacated niches. In our other work (Ma et al., 2017), we found that rapid redistribution of microglia from the inner to the outer retina in response to RPE injury creates empty niches in the inner retina which are then filled by infiltrating monocytes. It appears that there is a strong tendency of the retina to maintain myeloid cell homeostasis – replacement cells may be drawn from infiltrating monocytes if there is (1) an empty niche that is insufficiently filled by endogenous microglial proliferation and migration (Lund et al., 2018b), and/or (2) a viable route of vascular entry (i.e. such as through a compromised blood-CNS barrier). In TGFBR2 ablation, endogenous microglia proliferate and increase in number without presenting an obvious empty niche and without accompanying blood-retinal barrier breakdown. These factors may have limited the ability of monocytes to infiltrate the retina in this context. We have added to the Discussion:

“The reasons for why monocytic infiltration did not occur in this context are unclear and may be related to the absence of evident breakdown of the blood-retinal barrier, despite increased microglial activation, and the absence of an empty myeloid cell niche ready to accommodate infiltrating monocytes (Lund et al., 2018b). “(Discussion, p17).

The authors conclude that the "constitutive neuron-microglia interactions in the form of TGFβ are necessary in the maintenance of the orderly organization and trophic function of the microglial in the retina". What is the evidence that neurons are providing the TGFβ, how is it activated and which isoform of TGFβ is mediating the homeostatic suppression of microglial activation?

We appreciate the reviewer’s insightful question. While we have given it deep consideration, we may not be able to give it a full discussion in our manuscript. Here are some of our thoughts on these points:

On the topic of TGFβ ligand expression in the retina on the mRNA level, mRNA profiling of sorted retinal cell types in the adult mouse retina (Siegert et al., 2012) reveals that among the 3 Tgfb ligands, *Tgfb1* is primarily expressed by retinal microglia,*Tgfb2* is primarily expressed by a variety of retinal neurons, including photoreceptors, bipolar cells, amacrine cells, and retinal ganglion cells, and *Tgfb3* is expressed by amacrine neurons (see Author response image 2).

**Author response image 2. respfig2:** 

In addition, mRNA expression of *Tgfb1* and *Tgfb2* has been also attributed to Müller cells in culture (PMID: 9617551). On a protein level, TGFB2 has been found at higher levels in the monkey neural retina than TGFB1 on ELISA, with TGFB2 localizing to photoreceptors on immunohistochemistry (PMID: 7821377), but TGFB1 has also been localized to photoreceptors by immunohistochemistry in the human retina (Lutty et al., 1991).Immunohistochemical analyses of TGFB1, TGFB2, and TGFB3 have also been examined across different species (monkey, human, and cat) by Anderson et al., (PMID: Anderson et al., 1995) with overlapping patterns of immunopositivity in RPE cells, photoreceptors, Mueller cells, ganglion cells, hyalocytes, and vascular cells. In the context of glaucoma, TGFB2 has been located to astrocytes in the optic nerve head (PMID: 10396201) and TGFB1 to activated microglia there (PMID: 11391707).

Taken together, there is evidence that retinal neurons are a prominent source of TGFB2, and possibly some TGFB1 and TGFB3, while retinal microglia appear to express primarily TGFB1. While the details regarding the action of specific neuronal cell-type to microglia signaling via specific TGFb isoforms are not yet worked out, there is good evidence that retinal neurons are a prominent source of TGFb ligands (particularly TGFB2), with the possibility of microglia providing autocrine signaling via TGFB1. While we do not enter into these details in the manuscript, we have used the schematic in Figure 10 to display these likely relationships.

We have provided the following in the Discussion to reference the work examining the localization of TGFB ligands in the retina:

“TGFβ ligands, TGFβ1, TGFβ2, and TGFβ3, are expressed by multiple retinal cell types, including different classes of retinal neurons, endothelial cells, RPE cells, and retinal microglia (Anderson et al., 1995; Close et al., 2005; Lutty et al., 1993). In particular, Tgfb2 and Tgfb3 mRNA have been detected in amacrine, bipolar, and retinal ganglion cells (Siegert et al., 2012), and TGFB2 protein localized to photoreceptors (Lutty et al., 1991).”

The title is misleading. The data show that the absence of TGFβ signaling in microglial leads to exacerbated choroidal neovascularization, but the title suggests that this initiates the vessel growth.

This has been amended to read “Pathological transformation of retinal microglia in the absence of constitutive TGFβ signaling induces retinal degeneration and exacerbates choroidal neovascularization” as suggested by the reviewer.

Reviewer #3:[…] In many ways this paper is quite descriptive as there is no explanation for the neuronal loss following macrophage loss of TGFbR2. Nonetheless, the paper adds important new general information about macrophage neuronal interactions and TGFb control of macrophage function and related neuronal survival. I expect that with the techniques now available to look at multiple pathways the need to prove in detail that one pathway is involved after a genetic change in a cell will become less important.

We appreciate the reviewer’s comments and encouragements. We concur that there are likely multiple pathways that can contribute to microglial transitions from a healthy supportive state to one that contributes to pathological neurodegeneration. Even so, examining the specific contribution of TGF-b signaling to this transition is instructive and potentially valuable; because TGF-b has been highlighted in GWAS study as a contributor to AMD risk and has been put forward as a pathway to be targeted for AMD treatment, it may be insightful to discover further how this pathway connects to the physiology of retinal microglia.

We do not yet fully understand how neuronal loss in the retina occurs following microglial TGFBR2 ablation. We do not think that neurodegeneration arises from an insufficiency of microglial function, as prolonged microglial depletion in the retina, while resulting in synaptic degeneration, does not result in retinal thinning and atrophy (PMID: 28235894). As such, we hypothesize that neurodegeneration results from the acquisition of aberrant microglial functions that then induce neuronal cell death. We speculate that the secondary induction of Muller cell gliosis, which acquire a neurotoxic A2-like astrocytic signature (Liddelow et al., 2017), may contribute to the death of neurons.

Specific comments:Figure 1B: need to label y axis as Tgfbr2 and reference the control. Check all figures for more complete labeling with units and specific mRNAs etc.

The data in Figure 1B is referenced to Siegert et al., 2012 and additional information concerning the definition of expression level units and the linearity of the scale may be found in the originating paper. In the rest of the figures involving mRNA quantification that we have performed, we have checked to ensure that we consistently express it as a normalized value relative to the control group in the experiment.